# Is Plug-in Solver Sample-Efficient for Feature-based Reinforcement Learning?

**Qiwen Cui**
School of Mathematical Science,
Peking University
cuiqiwen@pku.edu.cn

**Lin F. Yang**
Electrical and Computer Engineering Department,
University of California, Los Angles
linyang@ee.ucla.edu

## Abstract

It is believed that a model-based approach for reinforcement learning (RL) is the key to reduce sample complexity. However, the understanding of the sample optimality of model-based RL is still largely missing, even for the linear case. This work considers sample complexity of finding an $\epsilon$-optimal policy in a Markov decision process (MDP) that admits a linear additive feature representation, given only access to a generative model. We solve this problem via a plug-in solver approach, which builds an empirical model and plans in this empirical model via an arbitrary plug-in solver. We prove that under the anchor-state assumption, which implies implicit non-negativity in the feature space, the minimax sample complexity of finding an $\epsilon$-optimal policy in a $\gamma$-discounted MDP is $O(K/(1-\gamma)^3\epsilon^2)$, which only depends on the dimensionality $K$ of the feature space and has no dependence on the state or action space. We further extend our results to a relaxed setting where anchor-states may not exist and show that a plug-in approach can be sample efficient as well, providing a flexible approach to design model-based algorithms for RL.

## 1 Introduction

Reinforcement learning (RL) [Sutton and Barto, 2018] is about learning to make optimal decisions in an unknown environment. It has been believed to be one of the key approaches to reach artificial general intelligence. In recent years, RL achieves phenomenal empirical successes in many real-world applications, e.g., game-AI [Vinyals et al., 2017], robot control [Duan et al., 2016], health-care [Li et al., 2018]. Most of these successful applications are based on a model-free approach, where the agent directly learns the value function of the environment. Despite superior performance, these algorithms usually take tremendous amount of samples. E.g. a typical model-free Atari-game agent takes about several hours of training data in order to perform well [Mnih et al., 2013]. Reducing sample complexity becomes a critical research topic of in RL.

It is well believed that model-based RL, where the agent learns the model of the environment and then performs planning in the model, is significantly more sample efficient than model-free RL. Recent empirical advances also justify such a belief (e.g. [Kaiser et al., 2019, Wang et al., 2019]). However, the understanding of model-based RL is still far from complete. E.g., how to deal with issues like model-bias and/or error compounding due to long horizon, in model-based RL is still an open question [Jiang et al., 2015], especially with the presence of a function approximator (e.g. a neural network) on the model. In order to get a better understanding on these issues, we target on the sample complexity question of model-based RL from a very basic setting: feature-based RL. In feature-based RL, we are given a hand-crafted or learned low-dimensional feature-vector for each state-action pair and the transition model can be represented by a linear combination of the feature vectors. Such a model recently has attracted much interest due to its provable guarantee

with model-free algorithms (e.g. [Yang and Wang, 2019, Jin et al., 2019]). In particular, we aim on answering the following fundamental question.

*Does a model-based approach on feature-based RL achieve near-optimal sample complexity?*

In particular, we focus on the generative model setting, where the agent is able to query samples freely from any chosen state-action pairs. Such a model is proposed by [Kearns and Singh, 1999, Kakade et al., 2003] and gains a great deal of interests recently [Azar et al., 2013, Sidford et al., 2018a, Yang and Wang, Zanette et al., 2019]. Moreover, we focus on the *plug-in solver approach*, which is probably the most intuitive and simplest approach for model-based RL: we first build an empirical model with an estimate of the transition probability matrix and then find a near optimal policy by planning in this empirical model via arbitrary plug-in solver. In the tabular setting, where the state and action spaces, $\mathcal{S}$ and $\mathcal{A}$, are finite, [Azar et al., 2013] shows that the *value estimation* of a plug-in approach is minimax optimal in samples. In particular, they show that to obtain an $\epsilon$-optimal value, the number of samples required to estimate the model is $\widetilde{O}(|\mathcal{S}||\mathcal{A}|/\epsilon^2(1-\gamma)^3)$.[1] Very recently, [Agarwal et al., 2019] proves that the *policy estimation* is also minimax optimal and with the same sample complexity. Unfortunately, these results cannot be applied to the function approximation setting, especially when the number of states becomes infinity.

In this paper, we show that the plug-in solver approach do achieve *near-optimal sample complexity* even in the feature-based setting, provided that the features are well conditioned. In particular, we show that under an anchor-state condition, where all features can be represented by the convex combination of some anchor-state features, an $\epsilon$-optimal policy can be obtained from an approximate model with only $\widetilde{O}(K/\epsilon^2(1-\gamma)^3)$ samples from the generative model, where $K$ is the feature dimension, independent of the size of state and action spaces. Under a more relaxed setting on the features, we also prove that finding an $\epsilon$-optimal policy only needs $\widetilde{O}(K \cdot poly(1/(1-\gamma))/\epsilon^2)$ samples. To achieve our results, we observe that the value function actually lies in an one-dimensional manifold and thus we can construct a series of auxiliary MDPs to approximate the value function. This auxiliary MDP technique breaks the statistical dependence that impedes the analysis. We have also extended our techniques to other settings e.g. finite horizon MDP (FHMDP) and two-players turn-based stochastic games (2-TBSG). To our best knowledge, this work first proves that plug-in approach is sample-optimal for feature-based RL and we hope our technique can boost analysis in broader settings.

## 2  Related Work

**Generative Model**   There is a line of research focusing on improving the sample complexity with a generative model, e.g. [Kearns and Singh, 1999, Kakade et al., 2003, Azar et al., 2012, 2013, Sidford et al., 2018a,b, Yang and Wang, Sidford et al., Zanette et al., 2019, Li et al., 2020, Zhang et al., 2020]. A classic algorithm under generative model setting is phased Q-learning [Kearns and Singh, 1999]. It uses $\widetilde{O}(|\mathcal{S}||\mathcal{A}|/\epsilon^2/poly(1-\gamma))$ samples to find an $\epsilon$-optimal policy, which is sublinear to the model size $|\mathcal{S}|^2|\mathcal{A}|$. Sample complexity lower bound for generative model has been established in [Azar et al., 2013, Yang and Wang, Sidford et al.]. In particular, [Azar et al., 2013] gives the first tight lower bound for unstructured discounted MDP. Later, this lower bound is generalized to feature-based MDP and two-players turn-based stochastic game in [Yang and Wang, Sidford et al.]. [Azar et al., 2013] also proves a minimax sample complexity $\widetilde{O}(|\mathcal{S}||\mathcal{A}|/\epsilon^2(1-\gamma)^3)$ of model-based algorithm for value estimation via the total variance technique. However, the sample complexity of value estimation and policy estimation differs in a factor of $\widetilde{O}(1/(1-\gamma)^2)$ [Singh and Yee, 1994]. The first minimax policy estimation result is given in [Sidford et al., 2018a], which proposes an model-free algorithm, known as Variance Reduced Q-value Iteration. This work has been extended to two-players turn-based stochastic game in [Sidford et al., Jia et al., 2019] and simultaneous game in [Zhang et al., 2020]. Recently, [Yang and Wang] develops an sample-optimal algorithm called Optimal Phased Parametric Q-Learning for feature-based RL. Their result requires $\epsilon \in (0, 1)$, while our result holds for $\epsilon \in (0, 1/\sqrt{1-\gamma})$. Plug-in solver approach is proved to be sample-optimal for tabular case in [Agarwal et al., 2019], which develops the absorbing MDP technique. However, their approach can not be generalized to linear transition model. A very recently paper [Li et al., 2020] develops a novel reward perturbation technique to remove the constraint on $\epsilon$ in tabular case.

Table 1: Sample complexity to compute $\epsilon$-optimal policy with generative model

| Algorithm | Sample Complexity | $\epsilon$-range | Problem type |
|---|---|---|---|
| Empirical QVI [Azar et al., 2013] | $\frac{\|\mathcal{S}\|\|\mathcal{A}\|}{(1-\gamma)^3\epsilon^2}$ | $(0, \frac{1}{\sqrt{(1-\gamma)\|\mathcal{S}\|}}]$ | Tabular MDP |
| Variance-reduced QVI [Sidford et al., 2018b] | $\frac{\|\mathcal{S}\|\|\mathcal{A}\|}{(1-\gamma)^3\epsilon^2}$ | $(0, 1]$ | Tabular MDP |
| Empirical MDP [Agarwal et al., 2019] | $\frac{\|\mathcal{S}\|\|\mathcal{A}\|}{(1-\gamma)^3\epsilon^2}$ | $(0, \frac{1}{\sqrt{1-\gamma}}]$ | Tabular MDP |
| Perturbed empirical MDP [Li et al., 2020] | $\frac{\|\mathcal{S}\|\|\mathcal{A}\|}{(1-\gamma)^3\epsilon^2}$ | $(0, \frac{1}{1-\gamma}]$ | Tabular MDP |
| QVI-MDVSS [Sidford et al.] | $\frac{\|\mathcal{S}\|\|\mathcal{A}\|}{(1-\gamma)^3\epsilon^2}$ | $(0, 1]$ | Tabular TBSG |
| OPPQ-Learning [Yang and Wang] | $\frac{K}{(1-\gamma)^3\epsilon^2}$ | $(0, 1]$ | Linear MDP |
| Two side PQ-Learning [Jia et al., 2019] | $\frac{KL^2}{(1-\gamma)^4\epsilon^2}$ | $(0, 1]$ | Linear TBSG |
| Empirical MDP (This work) | $\frac{K}{(1-\gamma)^3\epsilon^2}$ | $(0, \frac{1}{\sqrt{1-\gamma}}]$ | Linear MDP/TBSG |

Here $|\mathcal{S}|$ is the number of states, $|\mathcal{A}|$ is the number of actions, $\gamma$ is the discount factor, $K$ is the number of representative features, $\epsilon$ is the policy accuracy and $L$ is a coefficient measuring well-conditioned features.

**Function approximation RL**   Linear function approximation and linear transition model has been long studied [Bradtke and Barto, 1996, Melo and Ribeiro, 2007, Munos and Szepesvári, 2008, Chu et al., 2011, Abbasi-Yadkori et al., 2011, Jiang et al., 2015, Lever et al., 2016, Pires and Szepesvári, 2016, Azar et al., 2017, Jin et al., 2018, Lattimore and Szepesvari, 2019, Jin et al., 2019, Yang and Wang, Zanette et al., 2019, Du et al., 2019]. Linear function approximation algorithms have been developed in [Bradtke and Barto, 1996, Melo and Ribeiro, 2007, Du et al., 2019], which are also known as learning in the linear Q model. The concept of Bellman error in linear Q model is proposed in [Munos and Szepesvári, 2008], which obtains further analysis in [Jiang et al., 2017, 2015]. [Yang and Wang] proves that linear transition model is equivalent to linear Q model with zero Bellman error. A lot of work focuses on online learning setting, where the samples are collected by a learned policy. For tabular case, the tight regret lower bound is given in [Jaksch et al., 2010] and the tight regret upper bound is given in [Azar et al., 2017]. For linear feature RL, the best regret algorithms are given in [Jin et al., 2019] and [Yang and Wang, 2019], which study model-free algorithm and model-based algorithm respectively. However, the optimal regret for online linear RL is still unclear. For generative model setting, people care about sample complexity. Anchor-state assumption is the key to achieve minimax sample complexity, which is used in analyzing both the linear transition model and linear Q model [Yang and Wang, Zanette et al., 2019]. Similar concept known as soft state aggregation is developed in [Singh et al., 1995, Duan et al., 2019]. General function approximation has gathered increasing attention due to the impressive success of deep reinforcement learning. [Osband and Van Roy, 2014] gives the regret bound of a model-based algorithm with general function approximation and [Jiang et al., 2017, Sun et al., 2018] provides model-free algorithms and corresponding regret bounds. [Zanette et al., 2019] makes linear $Q^*$ assumption and their result holds only for $\|\lambda\|_1 \leq 1 + \frac{1}{H}$ so that the error will not amplify exponentially. Recently, [Wang et al., 2020] gives a efficient model-free algorithm without any structural assumption of the environment.

## 3   Preliminaries

In this section, we briefly introduce models that we will analyze in the following sections.

**Discounted Markov Decision Process**   A discounted Markov Decision Process (DMDP or MDP) is described by the tuple $M = (\mathcal{S}, \mathcal{A}, P, r, \gamma)$, where $\mathcal{S}$ and $\mathcal{A}$ are the state and action spaces, $P$ is the probability transition matrix which specifies the dynamics of the system, $r$ is the reward function of each state-action pair and $\gamma \in (0, 1)$ is the discount factor. Without loss of generality, we assume

$r(s,a) \in [0,1], \forall (s,a) \in (\mathcal{S}, \mathcal{A})$. The target of the agent is to find a stationary policy $\pi : \mathcal{S} \to \mathcal{A}$ that maximizes the discounted total reward from any initial state $s$:

$$V^\pi(s) := \mathbb{E} \left[ \sum_{t=0}^\infty \gamma^t r(s^t, \pi(s^t)) \,\middle|\, s^0 = s \right].$$

We call $V^\pi : \mathcal{S} \to \mathbb{R}$ the value function and for finite state space, it can be regarded as a $|\mathcal{S}|$-dimensional vector as well. It is well known that an optimal stationary policy $\pi^*$ exists and that it maximizes the value function for all states: $V^*(s) := V^{\pi^*}(s) = \max_\pi V^\pi(s), \forall s \in \mathcal{S}$. The action-value or Q-function of policy $\pi$ is defined as

$$Q^\pi(s,a) := \mathbb{E} \left[ r(s^0, a^0) + \sum_{t=1}^\infty \gamma^t r(s^t, \pi(s^t)) \,\middle|\, s^0 = s, a^0 = a \right] = r(s,a) + \gamma P(s,a) V^\pi,$$

where $P(s,a)$ is the $(s,a)$-th row of the transition matrix $P$. The optimal Q-function is denoted as $Q^* = Q^{\pi^*}$. Similarly, we have $Q^*(s,a) = \max_\pi Q^\pi(s,a), \forall (s,a) \in (\mathcal{S}, \mathcal{A})$. It is straightforward to show that $V^\pi(s) \in [0, \frac{1}{1-\gamma}]$ and $Q^\pi(s,a) \in [0, \frac{1}{1-\gamma}]$. Our target is to find an $\epsilon$-optimal policy $\pi$ such that $V^\pi(s) \leq V^*(s) \leq V^\pi(s) + \epsilon, \forall s \in \mathcal{S}$ for some $\epsilon > 0$.

**Feature-based Linear Transition Model**  We consider the case that the transition matrix $P$ has a linear structure. Suppose the learning agent is given a feature function $\phi : \mathcal{S} \times \mathcal{A} \to \mathbb{R}^K$:

$$\phi(s,a) = [\phi_1(s,a), \cdots, \phi_K(s,a)]$$

The feature function provides information about the transition matrix via a linear additive model.

**Definition 1.** (Feature-based Linear Transition Model) For a transition probability matrix $P$, we say that $P$ admits a linear feature representation $\phi$ if for every $s, a, s'$,

$$P(s'|s,a) = \sum_{k \in [K]} \phi_k(s,a) \psi_k(s'),$$

for some unknown functions $\psi_1, \cdots, \psi_K : \mathcal{S} \to \mathbb{R}$.

The linear transition model implies a low-rank factorization of the transition matrix $P = \Phi \Psi$ and one composite $\Phi$ is given as the features. In such a model, the number of unknown parameters is $K|\mathcal{S}|$, rather than $|\mathcal{S}|^2 |\mathcal{A}|$ for unstructured MDP. The linear transition model is closely related to another widely studied feature-based MDP, i.e. linear Q model [Bradtke and Barto, 1996].

**Generative Model Oracle**  Suppose we have access to a generative model which allows sampling from arbitrary state-action pair: $s' \sim P(\cdot|s,a)$. It is different from the online sampling oracle where a policy is used to collect data. To estimate the transition kernel $P$, we call the generative model $N$ times on each state-action pair in $\mathcal{K}$, where $|\mathcal{K}| = K$.[2] Then an estimate of the partial transition kernel $P_\mathcal{K}$ is:

$$\widehat{P}_\mathcal{K}(s'|s,a) = \frac{\text{count}(s,a,s')}{N}, \tag{1}$$

where $\text{count}(s,a,s')$ is the number of times the state $s'$ is sampled from $P(\cdot|s,a)$. The total sample size is $KN$. For tabular case, $\mathcal{K}$ is set to be $\mathcal{S} \times \mathcal{A}$ and $\widehat{P} = \widehat{P}_\mathcal{K}$ is the estimate of full transition kernel. In the linear transition model, we have $K \ll |\mathcal{S}||\mathcal{A}|$ so that the sample complexity is greatly reduced. The selection of $\mathcal{K}$ and the estimation of full transition kernel $P$ will be discussed in the next section.

**Plug-in Solver Approach**  In the empirical MDP $\widehat{M}$, we make use of a plug-in solver to get an approximately optimal policy. The plug-in solver receives $\widehat{M}$ with known transition distributions and outputs an $\epsilon_{PS}$-optimal policy $\widehat{\pi}$. Our goal is to prove that $\widehat{\pi}$ is also an approximately optimal policy in the true MDP $M$. In fact, we can assume $\epsilon_{PS} = 0$ as an MDP can be exactly solved in polynomial time. Even so, we consider the general case as reaching an approximately optimal policy is much less time-consuming than finding the exact optimal policy. We regard this as a tradeoff between time complexity and policy optimality. Approximate dynamic programming methods like LSVI/FQI [Kearns and Singh, 1999] can utilize the features to achieve $\widetilde{O}(\text{poly}(K(1-\gamma)^{-1}\epsilon^{-1}))$ computational complexity . In addition, learning algorithm 'Optimal Phased Parametric Q-Learning' in [Yang and Wang] can be used to do planning, which has computational complexity of $\widetilde{O}(K(1-\gamma)^{-3}\epsilon^{-2})$.

# 4 Empirical Model Construction

To construct an empirical MDP for linear transition model, the estimated transition kernel needs to be non-negative and sum to one. In our work, we propose a simple but effective method (Algorithm 1) to estimate the transition matrix.

**Proposition 1.** *Assume we have a linear transition model with feature function $\phi : \mathcal{S} \times \mathcal{A} \to \mathbb{R}^K$. For a row basis index set $\mathcal{K}$ of $\phi$, there exists $\{\lambda_k^{s,a}\}$ such that $\phi(s,a) = \sum_{k \in \mathcal{K}} \lambda_k^{s,a} \phi(s_k, a_k), \forall k \in \mathcal{K}, (s,a) \in (\mathcal{S}, \mathcal{A})$, Then $\widehat{P}(s'|s,a) = \sum_{k \in \mathcal{K}} \lambda_k^{s,a} \widehat{P}_{\mathcal{K}}(s'|s_k, a_k)$ satisfies*

1. *$\widehat{P}(s'|s,a)$ is an unbiased estimate of $P(s'|s,a)$,*

2. *$\sum_{k \in \mathcal{K}} \lambda_k^{s,a} = 1$ and $\sum_{s'} \widehat{P}(s'|s,a) = 1$,*

3. *if $\lambda_k^{s,a} \geq 0, \forall k \in \mathcal{K}, (s,a) \in (\mathcal{S}, \mathcal{A})$, then $\widehat{P}(s'|s,a) \geq 0, \forall (s,a,s') \in (\mathcal{S}, \mathcal{A}, \mathcal{S})$,*

*where $\widehat{P}_{\mathcal{K}}$ is given in (1).*

This proposition shows that the output of Algorithm 1 is an eligible estimate of the transition kernel $P$, when a proper state-action set $\mathcal{K}$ is chosen. Different choices of $\mathcal{K}$ can lead to different $\widehat{P}$. To ensure the non-negativity of $\widehat{P}$, we use a special class of state-action set $\mathcal{K}$.

**Assumption 1.** *(Anchor-state assumption) There exists a set of anchor state-action pairs $\mathcal{K}$ such that for any $(s,a) \in \mathcal{S} \times \mathcal{A}$, its feature vector can be represented as a convex combination of the anchors $\{(s_k, a_k)|k \in \mathcal{K}\}$:*

$$\exists \{\lambda_k^{s,a}\} : \phi(s,a) = \sum_{k \in \mathcal{K}} \lambda_k^{s,a} \phi(s_k.a_k), \sum_{k \in \mathcal{K}} \lambda_k^{s,a} = 1, \lambda_k \geq 0, \forall k \in \mathcal{K}, (s,a) \in (\mathcal{S}, \mathcal{A}).$$

Anchor-state assumption means the convex hull of feature vectors is a polyhedron with $|\mathcal{K}|$ nodes. Choosing these nodes to be the anchor-state set $\mathcal{K}$, the estimate given by Algorithm 1 is guaranteed to be a probability matrix, as a direct application of Proposition 1. This assumption has also been studied as soft state aggregation model [Singh et al., 1995, Duan et al., 2019]. Without loss of generality, we assume $\phi(s,a)$ to be a probability vector in the analysis, otherwise we can use $\{\lambda_k^{s,a}\}$ as the feature. The same anchor condition has been studied in [Yang and Wang] with a model-free algorithm. A modification of this condition has been proposed in [Zanette et al., 2019] to reach linear error propagation for value iteration.

**Proposition 2.** *If we have $N$ i.i.d. samples from each state-action pair in $\mathcal{K}$, then an unbiased estimate $\widehat{P}$ of the transition model is obtained from Algorithm 1:*

$$\widehat{P}(s'|s,a) = \sum_{k \in \mathcal{K}} \lambda_k^{s,a} \widehat{P}_{\mathcal{K}}(s'|s_k, a_k),$$

*where $\widehat{P}_{\mathcal{K}}(s'|s,a) = \frac{\text{count}(s,a,s')}{N}$ and $\{\lambda_k^{s,a}\}$ are coefficients defined in Proposition 1. In addition, if Assumption 1 holds, $\widehat{P}$ is a probability transition matrix.*

Proposition 2 shows that an empirical MDP $\widehat{M} = (\mathcal{S}, \mathcal{A}, \widehat{P}, r, \gamma)$ can be constructed by substituting the unknown transition matrix $P$ in $\mathcal{M}$ with $\widehat{P}$. The reward function $r$ is assumed to be known.[3] In

the following sections, we will analyze the property of $\widehat{M}$ and prove the concentration property of the optimal policy in $\widehat{M}$. We will use $\widehat{V}$ and $\widehat{\pi}$ to denote value function and policy in $\widehat{M}$.

---

**Algorithm 1:** Plug-in Solver Based Reinforcement Learning

---

**Input:** A generative model that can output samples from distribution $P(\cdot|s,a)$ for query $(s,a)$, a plug-in solver.
**Initial**: Sample size: $N$, state-action set $\mathcal{K}$;
**for** *(s,a) in $\mathcal{K}$* **do**
  | Collect $N$ samples from $P(\cdot|s,a)$;
**end**
Compute $\widehat{P}_\mathcal{K}(s'|s,a) = \frac{count(s,a,s')}{N}$;
Compute the linear combination coefficients $\lambda_k^{s,a}$ that satisfies $\phi(s,a) = \sum_{k\in\mathcal{K}} \lambda_k^{s,a}\phi(s_k.a_k)$;
Estimate transition distribution $\widehat{P}(s'|s,a) = \sum_{k\in\mathcal{K}} \lambda_k^{s,a}\widehat{P}_\mathcal{K}(s'|s_k,a_k)$;
Construct the empirical MDP: $\widehat{M} = (\mathcal{S},\mathcal{A},\widehat{P},r,\gamma)$ for DMDP, $\widehat{M} = (\mathcal{S},\mathcal{A},\widehat{P},r,H)$ for
  FHMDP, or $\widehat{M} = (\mathcal{S}_1.\mathcal{S}_2,\mathcal{A},\widehat{P},r,\gamma)$ for 2-TBSG;
Call the plug-in solver: input empirical model $\widehat{M}$ and output an $\epsilon_{\mathrm{PS}}$-optimal policy $\widehat{\pi}$ in $\widehat{M}$;
**Output:** $\widehat{\pi}$

---

## 5 Plug-in Solver Approach for Linear Transition Models

In this section, we analyze the sample complexity upper bounds for discounted MDP. Generally, we use Algorithm 1 to construct an empirical MDP $\widehat{M}$ and then use a plug-in solver to find an $\epsilon_{\mathrm{PS}}$-optimal policy $\widehat{\pi}$ in $\widehat{M}$. Algorithm 1 gives a formal framework of the plug-in solver approach. The target of our analysis is to prove that $\widehat{\pi}$ is an approximately optimal policy in the true MDP $M$.

Intuitively, $\widehat{P}$ is close to $P$ when it is constructed with sufficiently many samples, so $\widehat{M}$ is similar to $M$ and the optimal policy in $\widehat{M}$ is an approximately optimal policy in $M$. However, if we require that $\widehat{P}$ is close to $P$ in total variation distance, which is indeed a sufficient condition to obtain an approximately optimal policy for $M$, the number of samples needed is proportional to $K|\mathcal{S}|$, and hence is sample-inefficient. The same phenomena has been affecting the sample complexity in the tabular setting, [Azar et al., 2013], where their sample complexity of getting a constant optimal policy is at least $O(|\mathcal{S}|^2|\mathcal{A}|(1-\gamma)^{-3})$, which is at least the number of entries of the probability transition matrix. In the remaining part of this section, we leverage the sample de-coupling ideas from [Agarwal et al., 2019], variance preserving ideas from [Yang and Wang], and novel ideas to decouple MDP with linear feature representation to eventually establish our near-optimal sample complexity bound.

### 5.1 Linear Transition Model with Anchor State Assumption

In this section, we gives a minimax sample complexity of Algorithm 1 for feature-based MDP with anchor state assumption. The main results are shown below and then we introduce the auxiliary MDP technique, which is the key in the analysis.

**Theorem 1.** *(Sample complexity for DMDP) Suppose Assumption 1 is satisfied and the empirical model $\widehat{M}$ is constructed as in Algorithm 1. Set $\delta \in (0,1)$ and $\epsilon \in (0,(1-\gamma)^{-1/2}]$. Let $\widehat{\pi}$ be an $\epsilon_{\mathrm{PS}}$-optimal policy in $\widehat{M}$. If $N \geq \frac{c\log(cK(1-\gamma)^{-1}\delta^{-1})}{(1-\gamma)^3\epsilon^2}$, then with probability larger than $1-\delta$, we have*

$$Q^{\widehat{\pi}} \geq Q^* - \epsilon - \frac{3\epsilon_{\mathrm{PS}}}{1-\gamma},$$

*where $c$ is a constant.*

Theorem 1 shows that with $KN = \widetilde{O}(K/\epsilon^2(1-\gamma)^3)$ samples, an $\epsilon_{\mathrm{PS}}$-optimal policy in $\widehat{M}$ is an $\epsilon + 3\epsilon_{\mathrm{PS}}/(1-\gamma)$ policy in the true model with large probability. As we can solve $\widehat{M}$ to arbitrary accuracy (i.e. $\epsilon_{\mathrm{PS}} \to 0$) without collecting additional samples, this sample complexity matches the sample complexity lower bound given in [Yang and Wang]. We prove this theorem by using the auxiliary MDP technique to analyze the concentration property of $Q^{\widehat{\pi}^*} - \widehat{Q}^*$ and thus show

that three terms in $Q^* - Q^{\widehat{\pi}} = (Q^* - \widehat{Q}^{\pi^*}) + (\widehat{Q}^{\pi^*} - \widehat{Q}^*) + (\widehat{Q}^* - Q^{\widehat{\pi}})$ can be bounded. As we have $Q^\pi - \widehat{Q}^\pi = (I - \gamma P^\pi)^{-1}(P - \widehat{P})\widehat{V}^\pi$, which will be proved in the supplementary material, the main task in the analysis is to portray the concentration of $|(P - \widehat{P})\widehat{V}^{\pi^*}|$ and $|(P - \widehat{P})\widehat{V}^*|$. Due to the dependence between $\widehat{V}^{\pi^*}$, $\widehat{V}^*$ and $\widehat{P}$, conventional concentration arguments are not applicable. To decouple the dependence, we construct a series of auxiliary MDPs. In auxiliary models, transition distributions from all state-action pairs in $\mathcal{K}$ except a specific pair $(s, a)$ are equal to $\widehat{P}_{\mathcal{K}}$, while transition distribution from $(s, a)$ is $P(s, a)$. Then we prove that value function in $\widehat{M}$ can be approximated by tuning the reward in auxiliary model. Now, we give a rigorous definition for auxiliary MDP.

**Definition 2.** (Auxiliary DMDP) Suppose Assumption 1 holds. For an empirical MDP $\widehat{\mathcal{M}} = (\mathcal{S}, \mathcal{A}, \widehat{P} = \Phi\widehat{P}_K, r, \gamma)$ and a given state pair $(s, a) \in \mathcal{K}$, the auxiliary transition model is $\widetilde{\mathcal{M}}_{s,a,u} = (\mathcal{S}, \mathcal{A}, \widetilde{P} = \Phi\widetilde{P}_K, r + u\Phi^{s,a}, \gamma)$, where

$$\widetilde{P}_{\mathcal{K}}(s', a') = \begin{cases} \widehat{P}_{\mathcal{K}}(s', a') & if \ (s', a') \neq (s, a), \\ P(s, a) & otherwise. \end{cases}$$

$\Phi^{s,a} \in \mathbb{R}^{|\mathcal{S}||\mathcal{A}|}$ is the column vector of $\Phi$ that corresponds to $(s, a)$ in $\mathcal{K}$ and $u \in \mathbb{R}$ is a scalar.

As $\Phi$ is a probability transition matrix by Assumption 1, $\widetilde{P} = \Phi\widetilde{P}_{\mathcal{K}}$ is a probability transition matrix as well, so the auxiliary MDP is well defined. We use $\widetilde{V}_{s,a,u}$, $\widetilde{Q}_{s,a,u}$ and $\widetilde{\pi}_{s,a,u}$ to denote value function, action-value function and policy in $\widetilde{M}_{s,a,u}$, and we omit $(s, a)$ when there is no misunderstanding. We prove that there always exist a $u$ such that $\widehat{Q}^\pi = \widetilde{Q}^\pi_{s,a,u}$.

If we take $\widehat{Q}^\pi$ as a function of $\widehat{P}(s, a)$, then this function maps a $|\mathcal{S}| - 1$ dimensional probability simplex to $\mathbb{R}^{|\mathcal{S}||\mathcal{A}|}$. We observe a surprising fact that the range of $\widehat{Q}^\pi$ actually lies in a one dimensional manifold in $\mathbb{R}^{|\mathcal{S}||\mathcal{A}|}$, as tuning the scalar $u$ is sufficient to recover $\widehat{Q}^\pi$. Therefore, a one-dimensional $\epsilon$-net is sufficient to capture $\widehat{Q}^\pi$. By using evenly spaced points in the interval $U^\pi_{s,a}$, which is a bounded interval that contains all possible $u$, we show that the Q-function of corresponding auxiliary MDPs , $\widetilde{Q}_{s,a,u}$, forms the $\epsilon$-net. We prove that the auxiliary MDP is robust to coefficient $u$, and that an $\epsilon$-net on $u$ leads to an $\epsilon/(1 - \gamma)$-net on $\widetilde{Q}_{s,a,u}$. Given these properties of auxiliary DMDP, we can provide the proof sketch of Theorem 1. We denote a finite set $B^\pi_{s,a}$ to be made up of enough evenly spaced points in interval $U^\pi_{s,a}$. If $|B^\pi_{s,a}|$ is large enough, $B^\pi_{s,a}$ contains a $u'$ that is close enough to $u^\pi$. Thus $\widetilde{Q}^\pi_{u'}$ approximates $\widehat{Q}^\pi$. As $B^\pi_{s,a}$ has no dependence on $\widehat{P}(s, a)$, a union bound can be provided with Beinstein inequality and total variance technique [Azar et al., 2013, Yang and Wang, Agarwal et al., 2019]. Combining everything together, we can concentrate $|(P(s, a) - \widehat{P}(s, a))\widehat{V}^\pi|$ and hence finally bound $Q^* - Q^{\widehat{\pi}}$.

The next result is about the sample complexity when model misspecification exists. Model misspecification means that the transition matrix cannot be fully expressed by the linear combination of features, i.e. $P = \Phi\bar{P}_\mathcal{K} + \Xi$, where $\Xi$ is the approximation error or noise of linear transition model and $\Phi\bar{P}_\mathcal{K}$ is the underlying true transition. In this condition, the estimate $\widehat{P}$ given by Algorithm 1 can be biased, and the degree of perturbation depends on $\xi = \|\Xi\|_\infty$.

**Theorem 2.** (Sample complexity for DMDP with model misspecification) Suppose $M = (\mathcal{S}, \mathcal{A}, P, r, \gamma)$ has an approximate linear transition model $\bar{P}$ such that $\bar{P}$ admits a linear feature representation $\phi$ and there exists some $\xi \geq 0$ that $\|P(\cdot|s, a) - \bar{P}(\cdot|s, a)\|_{TV} \leq \xi, \forall (s, a) \in (\mathcal{S}, \mathcal{A})$. Suppose Assumption 1 is satisfied and the empirical model $\widehat{M}$ is constructed as in Algorithm 1. Set $\delta \in (0, 1)$ and $\epsilon \in (0, (1 - \gamma)^{-1/2}]$. Let $\widehat{\pi}$ be an $\epsilon_{PS}$-optimal policy for $\widehat{M}$. If $N \geq \frac{c \log(cK(1-\gamma)^{-1}\delta^{-1})}{(1-\gamma)^3\epsilon^2}$, then with probability larger than $1 - \delta$,

$$Q^{\widehat{\pi}} \geq Q^* - \epsilon - \frac{3\epsilon_{PS}}{1 - \gamma} - \frac{16\sqrt{\xi}}{(1 - \gamma)^2},$$

where $c$ is a constant.

Theorem 2 implies that for an approximately linear transition model, the suboptimality of $\widehat{\pi}$ only increase $\frac{16\sqrt{\xi}}{(1-\gamma)^2}$. The proof is given in the supplementary material and the high level idea is to separate the linear model part and perturbation part in the empirical model.

## 5.2 General Linear Transition Model

If the anchor-state assumption is not satisfied, the estimate $\widehat{P}$ given by Algorithm 1 is not guaranteed to be a non-negative matrix, and thus not a probability transition matrix. Next proposition shows that even if the features approximately satisfy the anchor state assumption, the estimated transition $\widehat{P}$ is still not a probability transition matrix with high probability.

**Proposition 3.** *Suppose the samples are from state-action pairs $\mathcal{K}$ and the unbiased estimate of the transition probability matrix given by Algorithm 1 is $\widehat{P}$. If $\max_{s,a} \sum_{k \in \mathcal{K}} |\lambda_k^{s,a}| = L > 1$ and $K \geq 2$, where $\{\lambda_k^{s,a}\}$ is defined as in Proposition 1, there exists true model $P$ such that with probability larger than $1/3$, $\widehat{P}$ is not a probability transition matrix for sample size $N \geq C$, where $C$ is a constant.*

Note that we have $\max_{s,a} \sum_{k \in \mathcal{K}} \lambda_k^{s,a} = 1$, so $L \geq 1$ is always satisfied and $L = 1$ is equivalent to anchor state assumption. Proposition 3 shows that without anchor state assumption, we can always find an MDP $M$ such that the empirical MDP $\widehat{M}$ is not a well-defined MDP with large probability. The estimated transition distributions $\widehat{P}$ sum to one, but are not necessarily non-negative. This kind of MDP is known as pseudo MDP [Yao et al., 2014] and [Lever et al., 2016, Pires and Szepesvári, 2016, Lattimore and Szepesvari, 2019] analyze the error bound induced under unrealizability. In a pseudo MDP, well-known properties of bellman operator like contraction and monotonicity no longer exist, so there is no optimal value/policy. An example of pseudo MDP without optimal value/policy is provided in the supplementary material. However, plug-in solvers like value iteration still work in pseudo MDP. In this part, we give a sample complexity bound for value iteration solver. To facilitate analysis, we need a regularity assumption on features.

**Assumption 2.** *(Representative States and Regularity of Features) There exists a set of state-action pairs $\mathcal{K}$ and a scalar $L \geq 1$ such that for any $(s,a) \in \mathcal{S} \times \mathcal{A}$, its feature vector can be represented as a linear combination of the state-actions $\{(s_k, a_k) | k \in \mathcal{K}\}$:*

$$\exists \{\lambda_k^{s,a}\} : \phi(s,a) = \sum_{k \in \mathcal{K}} \lambda_k^{s,a} \phi(s_k.a_k), \sum_{k \in \mathcal{K}} |\lambda_k^{s,a}| \leq L, \forall (s,a) \in (\mathcal{S}, \mathcal{A}).$$

This assumption means that the selected state-action pairs $\mathcal{K}$ can represent all features by linear combinations with bounded coefficients, which avoids error explosion in the iterative algorithm. In theorem 3, we give the sample complexity for value iteration solver and the proof is provided in the supplementary material.

**Theorem 3.** *(Value iteration solver for general linear MDP) Suppose Assumption 2 is satisfied and the empirical model $\widehat{M}$ is constructed as in Algorithm 1. Set $\delta \in (0,1)$ and $\epsilon \in (0, 1/(1-\gamma))$. $\widehat{V}$ is the value function of applying value iteration for $O(1/(1-\gamma))$ times in the empirical MDP $\widehat{M} = (\mathcal{S}, \mathcal{A}, \widehat{P}, r, \gamma)$. Let $\widehat{\pi}$ be the greedy policy with respect to $\widehat{V}$. If $N \geq \frac{cL^2 \log(cK(1-\gamma)^{-1}\delta^{-1})}{\epsilon^2} \text{poly}(1/(1-\gamma))$, then with probability larger than $1-\delta$,*

$$Q^{\widehat{\pi}} \geq Q^* - \epsilon,$$

*where c is a constant.*

This theorem shows that without anchor-state assumption, value iteration solver is still a sample efficient algorithm. For pseudo MDP, the optimal policy/value no longer exist, so different plug-in solver may output significantly different policies. Even though we can assume all solvers are time efficient, an one to one analysis is needed and we leave this as future work.

## 6 Extensions to Finite Horizon MDP and Two-players Turn-based Stochastic Game

Our approach for discounted MDP can be extended to finite horizon MDP (FHMDP) and two-players turn-based stochastic game (2-TBSG). Due to space limitation, preliminary about FHMDP and

2-TBSG is given in the supplementary material. All of the three decision models use a transition probability matrix to represent the system dynamic, so the framework of plug-in solver approach is the same (i.e. Algorithm 1). Here we directly present the sample complexity results here.

**Theorem 4.** *(Sample complexity for FHMDP) Suppose Assumption 1 is satisfied and the empirical model $\widehat{M}$ is constructed as in Algorithm 1. Set $\delta \in (0,1)$ and $\epsilon \in (0, (1-\gamma)^{-1/2}]$. Let $\widehat{\pi}$ be any $\epsilon_{\mathrm{PS}}$-optimal policy for $\widehat{\mathcal{M}}$. If $N \geq \frac{c \log(cKH\delta^{-1})H^3 \min\{|\mathcal{S}|, K, H\}}{\epsilon^2}$, then with probability larger than $1-\delta$, $Q^{\widehat{\pi}} \geq Q^* - \epsilon - 3\epsilon_{\mathrm{PS}}H$, where $c$ is a constant.*

**Theorem 5.** *(Sample complexity for 2-TBSG) Suppose Assumption 1 is satisfied and the empirical model $\widehat{M}$ is constructed as in Algorithm 1. Set $\delta \in (0,1)$ and $\epsilon \in (0, (1-\gamma)^{-1/2}]$. Let $\widehat{\pi} = (\widehat{\pi}_1, \widehat{\pi}_2)$ be any $\epsilon_{\mathrm{PS}}$-optimal policy for $\widehat{\mathcal{M}}$. If $N \geq \frac{c \log(cK(1-\gamma)^{-1}\delta^{-1})}{(1-\gamma)^3 \epsilon^2}$, then with probability larger than $1-\delta$, $|Q^{\widehat{\pi}} - Q^*| \leq \epsilon + \frac{3\epsilon_{\mathrm{PS}}}{1-\gamma}$, where $c$ is a constant.*

The above theorems indicate that plug-in solver is sample efficient for FHMDP and 2-TBSG. The high level idea of the proof is similar to the analysis of discounted MDP, i.e. we use a series of auxiliary MDP to break the statistical dependence. For FHMDP, the rewards in auxiliary MDPs need to be tuned step by step, which leads to the extra $H$ dependency. For 2-TBSG, we analyze the counter policy, or known as the best response policy, in true model and in empirical model. The details are problem-specific and due to space limitation, we provide them in the supplementary material.

## 7 Discussion

This paper studies the sample complexity of plug-in solver approach in feature-based MDPs, including discounted MDP, finite horizon MDP and stochastic games. We tackle a basic and important problem in reinforcement learning: whether planning in an empirical model is sample efficient to give an approximately optimal policy in the real model. To our best knowledge, this is the first result proving minimax sample complexity for the plug-in solver approach in feature-based MDPs. We hope that the new technique in our work can be reused in more general settings and motivate breakthroughs in other domains.

Our work also opens up several directions for future research, which is listed below.

- *Improve the dependence on $H$ for finite horizon MDP.* The sample complexity we give for finite horizon MDP is $\widetilde{O}(KH^4\epsilon^{-2})$, which has an extra $H$ dependence compared with the discounted case. We conjecture that the plug-in solver approach should enjoy the optimal $\widetilde{O}(KH^3\epsilon^{-2})$ complexity as model-free algorithms [Yang and Wang]. This may require new techniques as absorbing MDP is not well suitable for finite horizon MDP.

- *Improve the result for stochastic game.* Our result for turn-based stochastic game is for finding a strategy with value $\epsilon$-close to the Nash equilibrium value, while the final objective is to find an $\epsilon$-Nash equilibrium value (for definition see [Sidford et al.]). We need a more refined analysis to tackle the complex dependence between two players.

- *Extend the range of $\epsilon$.* Our result holds for $\epsilon \in (0, \frac{1}{\sqrt{1-\gamma}}]$, which is better than previous model-free result [Yang and Wang]. Recently, [Li et al., 2020] develops a novel reward perturbation technique to prove that the minimax sample complexity holds for $\epsilon \in (0, \frac{1}{1-\gamma}]$. A direct application of their result would lead to $\log(|\mathcal{S}||\mathcal{A}|)$ factor in linear MDP. The full $\epsilon$ range for linear MDP is still an open problem.

- *Beyond linear MDP.* Currently most works focus on tabular and linear setting with generative model, which provide intuition for solving the ultimate setting of general function approximation like neural network. One interesting problem is whether we can develop provably efficient model-based algorithm under general function approximation setting, as the construction of the empirical model seems to be difficult even for linear Q-function assumption.

## Broader Impact

We believe that our work can benefit both theory and algorithm researches of model-based reinforcement learning (MBRL), as the sample efficiency of MBRL has been long observed but lack theoretical analysis. Previously few sample complexity results for MBRL in feature-based MDP exist, so people may wonder if the gap between theory and application is treatable. Our work answers this question positively by theoretically proving the simplest MBRL method, which is the plug-in solver approach, can reach the minimax sample complexity. Researchers that are interested in MBRL theory can benefit from our results and techniques. In addition, we give a thorough analysis on linear features and show that anchor-state condition can measure the quality of features. This result can guide reinforcement learning practioners to design features and sample efficient algorithms. We believe that our research has no negative societal effects.

## Acknowledgements

We thank all anonymous reviewers for their insightful comments.

## Footnotes

[1] In $\widetilde{O}(f)$, $\log f$ factors are ignored.

[2]Our result holds for $|\mathcal{K}| = \widetilde{O}(K)$.

[3]If $r$ is unknown, we can assume it has the same structure as the transition matrix $p$ and estimate it in the same manner. With simple modification of our proof, we can show that only $O(K/\epsilon^2(1-\gamma)^2)$ is needed to get a sufficiently accurate estimation of $r$.

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
