[Supplementary Material]

# Is Plug-in Solver Sample Efficient for Feature-based Reinfocement Learning?

**Notations**   We use $P = \Phi\Psi$ to represent the linear transition model, where $\Phi \in \mathbb{R}^{|\mathcal{S}||\mathcal{A}|\times K}$ is the feature matrix consisting of feature vectors $\phi(s,a)$ and $\Psi$ is the corresponding unknown matrix from $\psi$. Proposition 1 means that $\Phi$ can be factorized as $\Phi = \Lambda\Phi_{\mathcal{K}}$ where $\Lambda \in \mathbb{R}^{|\mathcal{S}||\mathcal{A}|\times K}$ is the matrix of $\lambda_k^{s,a}$ and $\Phi_{\mathcal{K}}$ is the submatrix of $\Phi$. Thus we have $P = \Lambda\Phi_{\mathcal{K}}\Psi = \Lambda P_{\mathcal{K}}$. In addition, if we have feature $\Phi$ and state-action pairs $\mathcal{K}$, we can compute $\Lambda$ and $\Lambda$ can also be regarded as feature. Thus, we will use $\Phi$ and $\phi$ to represent $\Lambda$ and $\lambda$ in Appendix C and Appendix D.

We use $P^\pi \in \mathbb{R}^{|\mathcal{S}||\mathcal{A}|\times|\mathcal{S}||\mathcal{A}|}$ to denote the transition distributions on state action pairs that are induced by policy $\pi$. $P^* \in \mathbb{R}^{|\mathcal{S}||\mathcal{A}|\times|\mathcal{S}||\mathcal{A}|}$ is the transition matrix induced by optimal policy $\pi^*$. $P_{\mathcal{K}} \in \mathbb{R}^{|\mathcal{K}|\times|\mathcal{S}|}$ is the submatrix of $P$ consisting of rows in $\mathcal{K}$. $\widehat{P}^\pi, \widehat{P}^*, \widehat{P}_{\mathcal{K}}, \bar{P}_{\mathcal{K}}, \widetilde{P}^\pi, \widetilde{P}^*, \widetilde{P}_{\mathcal{K}}$ are defined in a similar manner. We use $P(s,a)$ to denote the row vector of $P$ that corresponds to $(s,a)$.

For a vector $V$, we use $V^2$, $|V|$, $\sqrt{V}$ and $<$ to denote the component-wise square, absolute value, square root and less-than. We define $Var_{s,a}(V) := P(s,a)V^2 - (P(s,a)V)^2$, $Var_{\mathcal{K}}(V) = [Var_{s_1,a_1}(V),\cdots,Var_{s_K,a_K}] \in \mathbb{R}^K$ and $Var_P(V) \in \mathbb{R}^{|\mathcal{S}||\mathcal{A}|}$ to consist of all $Var_{s,a}(V)$. A detailed description is provided in [1]. We use $\mathbf{1}$ to denote a column vector with all components to be 1. We use $[H]$ to denote $\{0, 1, \cdots, H-1\}$.

## A   Additional Preliminary

**Finite Horizon Markov Decision Process**   A Finite Horizon Markov decision process (FHMDP) is described by the tuple $\mathcal{M} = (\mathcal{S}, \mathcal{A}, P, r, H)$, which differs from DMDP only in that the discount factor $\gamma$ is replaced by the horizon $H$. The value function of policy $\pi$ is defined as:

$$V_h^\pi(s) := \mathbb{E}\left[\sum_{t=h}^{H-1} r(s^t, \pi_t(s^t))|s^h = s\right],$$

which depends on state $s$ and time step $h$. The target of the agent is to learn a policy $\pi = (\pi_0, \cdots, \pi_{H-1}), \pi_h : \mathcal{S} \to \mathcal{A}, \forall h \in [H]$ that maximize the total reward $V_0^\pi(s)$ from any initial state $s$. It is well known that the optimal policy for FHMDP maximizes the value function in each time step:

$$V_h^*(s) := V_h^{\pi^*}(s) = \max_\pi V^\pi(s), \forall s \in \mathcal{S}, h \in [H].$$

The action-value or Q-function $Q_h^\pi$ and optimal Q-function $Q^*$ are defined similarly as in DMDP. The relation between Q-function and value function is:

$$V_h^\pi(s) = Q_h^\pi(s, \pi_h(s)), \quad Q_h^\pi(s,a) = r(s,a) + \gamma P(s,a)V_{h+1}^\pi.$$

If the value function of a policy $\pi$ is $\epsilon$-close to the optimal value function in all time steps, the policy is called a $\epsilon$-optimal policy:

$$Q_h^*(s,a) \geq Q_h^\pi(s,a) \geq Q_h^*(s,a) - \epsilon, \quad \forall (s,a), h \in [H].$$

Without loss of generality, we assume $r \in [0, 1]$ and then we have $V_h^\pi \in [0, H - h]$. Note that in a normal FHMDP, the reward in each time step is identical, but in our analysis, we will construct FHMDPs with different rewards in each step.

**Two Player Turn-based Stochastic Game** A discounted turn-based two-player zero-sum stochastic games (2-TBSG) is described as the tuple $\mathcal{G} = (\mathcal{S}_{max}, \mathcal{S}_{min}, \mathcal{A}, P, r, \gamma)$. It is a generalized version of DMDP which includes two players competing with each other. Player 1 aims to maximize the total reward with policy $\pi_1$ while player 2 aims to minimize it with policy $\pi_2$. We denote policy $\pi := (\pi_1, \pi_2)$ to be the overall policy. Given policy $\pi$, the value function and Q-function can be defined as in DMDP.

From the perspective of player 1, if the policy of player 2 $\pi_2$ is given, the 2-TBSG degenerates to a DMDP, so the optimal policy exists for player 1. This optimal policy depends on $\pi_2$, so we call it the counter policy to $\pi_2$ and denote it as $c_1(\pi_2)$. Similarly we can define $c_2(\pi_1)$ as the counter policy of $\pi_1$ for player 2. For simplicity, we ignore the subscript in $c_1$ and $c_2$ when it is clear in the context.

To solve a 2-TBSG, our goal is to find the Nash equilibrium policy $\pi^* = (\pi_1^*, \pi_2^*)$, where $\pi_1^* = c(\pi_2^*), \pi_2^* = c(\pi_1^*)$. For this policy, neither player can benefit from change its policy alone.

We give the following well-known properties of 2-TBSG without proof (see. e.g. [4]), which can be regarded as generalized optimality property in DMDP.

- $V^{c(\pi_2),\pi_2} = \max_{\pi_1} V^{\pi_1,\pi_2}$
- $V^{\pi_1,c(\pi_1)} = \max_{\pi_2} V^{\pi_1,\pi_2}$
- $V^{\pi_1^*,\pi_2^*} = \min_{\pi_2} V^{c(\pi_2),\pi_2}$
- $V^{\pi_1^*,\pi_2^*} = \max_{\pi_1} V^{\pi_1,c(\pi_1)}$

Our target is to find an $\epsilon$-optimal policy $\pi = (\pi_1, \pi_2)$ such that $\left|Q^\pi(s,a) - Q^*(s,a)\right| \leq \epsilon, \forall(s,a)$ for some $\epsilon > 0$.

# B   Empirical Model Construction

In this section, we prove Proposition 1, Proposition 2 and Proposition 3, which give the theoretical guarantees of our model construction algorithm (Algorithm 1).

*Proof of Proposition 1.* Here we prove the three arguments in Proposition 1.

1. As $\widehat{P}_{\mathcal{K}}(s'|s_k, a_k) = \frac{\text{count}(s_k, a_k, s')}{N}$, $\widehat{P}_{\mathcal{K}}$ is an unbiased estimate of $P_{\mathcal{K}}$. Hence we have
$$\mathbb{E}\widehat{P}(s'|s,a) = \mathbb{E}\sum_{k\in\mathcal{K}}\lambda_k^{s,a}\widehat{P}_{\mathcal{K}}(s'|s_k,a_k) = \sum_{k\in\mathcal{K}}\lambda_k^{s,a}P(s'|s_k,a_k) = P(s'|s,a),$$
   where the last equality is from $P = \Lambda P_{\mathcal{K}}$.

2. We have that
$$\begin{aligned}\sum_{k\in\mathcal{K}}\lambda_k^{s,a} &= \sum_{k\in\mathcal{K}}\sum_{s'\in\mathcal{S}}\lambda_k^{s,a}P(s'|s_k,a_k)\\ &= \sum_{s'\in\mathcal{S}}\sum_{k\in\mathcal{K}}\lambda_k^{s,a}P(s'|s_k,a_k)\\ &= \sum_{s'\in\mathcal{S}}P(s'|s,a)\\ &= 1.\end{aligned}$$
   Thus we have $\sum_{k\in\mathcal{K}}\lambda_k^{s,a} = 1$. Then we have
$$\begin{aligned}\sum_{s'\in\mathcal{S}}\widehat{P}(s'|s,a) &= \sum_{s'\in\mathcal{S}}\sum_{k\in\mathcal{K}}\lambda_k^{s,a}\widehat{P}_{\mathcal{K}}(s'|s_k,a_k)\\ &= \sum_{k\in\mathcal{K}}\sum_{s'\in\mathcal{S}}\lambda_k^{s,a}\widehat{P}_{\mathcal{K}}(s'|s_k,a_k)\\ &= \sum_{k\in\mathcal{K}}\lambda_k^{s,a}\\ &= 1.\end{aligned}$$

3. If $\lambda_k^{s,a} \geq 0, \forall k \in \mathcal{K}, (s,a) \in (\mathcal{S}, \mathcal{A})$, then we have

$$\widehat{P}(s'|s,a) = \sum_{k \in \mathcal{K}} \lambda_k^{s,a} \widehat{P}_{\mathcal{K}}(s'|s_k, a_k) \geq 0,$$

as every component is non-negative.

$\square$

**Remark 1.** *Proposition 1 implies that when anchor-state condition is satisfied, $\Lambda$ is a probability matrix. Thus $P = \Lambda P_{\mathcal{K}}$ is factorized as a probability matrix into two probability transition matrix.*

*Proof of Proposition 2.* Combining Proposition 1 and Assumption 1, we directly have that $\sum_{s' \in \mathcal{S}} \widehat{P}(s'|s,a) = 1, \widehat{P}(s'|s,a) \geq 0$ for all $(s',s,a)$. Therefore $\widehat{P}$ is an eligible transition kernel. With Assumption 1 or not, we always have $\mathbb{E}\widehat{P}(s'|s,a) = P(s'|s,a)$, which means $\widehat{P}$ is an unbiased estimate of $P$. $\square$

*Proof of Proposition 3.* We consider the following case. Suppose $\mathcal{K} = \{(s_1,a_1), \cdots, (s_K,a_K)\}$. For a specific $(s,a) \notin \mathcal{K}$, we have $\lambda_{s_1,a_1}^{s,a} = \frac{1+L}{2}, \lambda_{s_2,a_2}^{s,a} = \frac{1-L}{2}$ and $\lambda_{s_k,a_k}^{s,a} = 0$ for $k \neq 1, 2$. For all other $(s',a') \neq (s,a)$, we set

$$\lambda_{s'',a''}^{s',a'} = \begin{cases} \mathbf{1}_{\{(s'',a'')=(s',a')\}} & \text{if } (s',a') \in \mathcal{K} \\ \frac{1}{K} & \text{otherwise} \end{cases}$$

It is easy to check that $\{\lambda_k^{s,a}\}$ is valid (We can set $\Phi = \Lambda$ and with state-action set $\mathcal{K}$, then the corresponding linear combination coefficients are $\Lambda$ as $\Phi_{\mathcal{K}} = I$). We set $P(s_1|s_1,a_1) = \frac{L-1}{L+1}, P(s_2|s_1,a_1) = \frac{2}{L+1}$ and $P(s_1|s_2,a_2) = 1$. Other transition distribution can be defined arbitrarily to construct an eligible $P$.

As $\text{count}(s_1,a_1,s_1)$ follows a binomial distribution with $p = P(s_1|s_1,a_1) = \frac{L-1}{L+1}$ and $\text{count}(s_2,a_2,s_1) = N$ as $P(s_1|s_2,a_2) = 1$, the estimate of $P(s_1|s,a) = \sum_{k \in \mathcal{K}} \lambda_k^{s,a} P(s_1|s_k,a_k) = 0$ is

$$\begin{aligned} \widehat{P}(s_1|s,a) &= \sum_{k \in \mathcal{K}} \lambda_k^{s,a} \widehat{P}(s_1|s_k,a_k) \\ &= \lambda_{s_1,a_1}^{s,a} \frac{\text{count}(s_1,a_1,s_1)}{N} + \lambda_{s_2,a_2}^{s,a} \frac{\text{count}(s_1,a_1,s_1)}{N} \\ &= \frac{L+1}{2} \frac{\text{count}(s_1,a_1,s_1)}{N} - \frac{L-1}{2}. \end{aligned}$$

$\widehat{P}(s_1|s,a)$ is a translated and scaled binomial distribution with zero mean. By central-limit theorem, $\sqrt{N}\widehat{P}(s_1|s,a)$ converges to Gaussian distribution with zero mean. Therefore, there exists constant $C$ that if $N \geq C$, we have $\mathbb{P}(\widehat{P}(s_1|s,a) \leq 0) \geq \frac{1}{3}$. This means the estimate $\widehat{P}$ is not a probability transition matrix with probability larger than 1/3. $\square$

**Remark 2.** *This example can be generalized to prove that $\widehat{P}$ is not non-negative with probability larger than $1 - 1/3^{[K/2]}$, by choosing $[K/2]$ such $s, a$ pairs and each pair independently leads to an ineligible estimate with probability 1/3.*

Now we give an example of pseudo MDP to show that the optimality in normal MDP no longer exists, which means that there is no policy $\pi^*$ such that $V^{\pi^*}(s) = \max_\pi V^\pi(s)$ for all $s \in \mathcal{S}$. Note that the Bellman operator $\mathcal{T}^\pi[Q] = r + \gamma P^\pi Q$ may still be an contraction if $\|P^\pi\|_{1,\infty} \leq \frac{1}{\gamma}$, but the monotonicity no longer exist, which means we do not have $\mathcal{T}^\pi[Q_1] \geq \mathcal{T}^\pi[Q_2]$ if $Q_1 \geq Q_2$.

We construct a pseudo MDP with $\mathcal{S} = \{s_1, s_2\}, \mathcal{A} = \{a_1, a_2\}$. The transition distributions and rewards are

$$P(s_1,a_1) = [0,1], P(s_1,a_2) = [0,1], P(s_2,a_1) = [1,0], P(s_2,a_2) = [-0.1, 1.1],$$

$$r(s_1,a_1) = 1, r(s_1,a_2) = 0, r(s_2,a_1) = 0, r(s_2,a_2) = 1.$$

In this pseudo MDP, there are four policies $\pi_1, \pi_2, \pi_3, \pi_4$, which correspond to choosing action $(a_1, a_1), (a_1, a_2), (a_2, a_1), (a_2, a_2)$ in state $(s_1, s_2)$. We use $V_1, V_2, V_3, V_4 \in \mathbb{R}^2$ to denote the corresponding value functions. Using Bellman equation, we have that

$$V_1 = [\frac{1}{1-\gamma^2}, \frac{\gamma}{1-\gamma^2}], V_2 = [0, 0], V_3 = [\frac{1}{1-\gamma}, \frac{1}{1-\gamma}],$$

$$V_4 = [\frac{\gamma}{0.1\gamma^2 - 1.1\gamma + 1}, \frac{1}{0.1\gamma^2 - 1.1\gamma + 1}].$$

With some calculation, we have that

$$\operatorname{argmax}_i V_i(s_1) = 2, \operatorname{argmax}_i V_i(s_2) = 3, \text{if } \gamma \leq \frac{10}{11}.$$

Therefore, in this pseudo MDP, no policy can achieve optimality in all states and no solver can output an $\epsilon$-optimal policy for arbitrary small $\epsilon$.

## C  Linear Transition Model with Anchor State Assumption

### C.1  Sample Complexity for Discounted MDP

Here we give the formal proof of Theorem 1. First we give the definition of $U_{s,a}^\pi$ and $U_{s,a}^*$, which is the set that contains all possible $u$ for different policies in auxiliary MDP.

**Definition 1.** (Feasible Set for $u$) For the auxiliary transition model $\widetilde{M}_{s,a,u} = (\mathcal{S}, \mathcal{A}, \widetilde{P} = \Phi \widetilde{P}_\mathcal{K}, r + u\Phi^{s,a}, \gamma)$, $U_{s,a}^\pi$ is defined as the set of $u$ such that $\widetilde{V}_u^\pi \in [0, 1/(1-\gamma)]^\mathcal{S}$, and $U_{s,a}^*$ is defined as the set of $u$ such that $\widetilde{V}_u^* \in [0, 1/(1-\gamma)]^\mathcal{S}$.

**Remark 3.** *Obviously, $u$ that satisfies $0 \leq r + u\Phi^{s,a} \leq 1$ is in both $U_{s,a}^\pi$ for arbitrary $\pi$ and $U_{s,a}^*$. Immediately, we have $0 \in U_{s,a}^\pi$ for arbitrary $\pi$ and $0 \in U_{s,a}^*$. Note that both $U_{s,a}^\pi$ and $U_{s,a}^*$ are independent of $\widehat{P}(s,a)$ and they are bounded intervals.*

**Notations**  We use $\widetilde{V}_{s,a,u}$, $\widetilde{Q}_{s,a,u}$ and $\widetilde{\pi}_{s,a,u}$ to denote value function, action value function and policy in $\widetilde{M}_{s,a,u}$, and we omit $(s,a)$ when there is no misunderstanding.

**Lemma 1.** *For an $\epsilon_{\mathrm{PS}}$-optimal policy $\widehat{\pi}$ in $\widehat{\mathcal{M}}$, we have*

$$\left\| Q^* - Q^{\widehat{\pi}} \right\|_\infty \leq \left\| Q^* - \widehat{Q}^{\pi^*} \right\|_\infty + \left\| \widehat{Q}^{\widehat{\pi}} - Q^{\widehat{\pi}} \right\|_\infty + \epsilon_{\mathrm{PS}}.$$

*Proof.* By the definition of $V^*$, we have

$$0 < Q^* - Q^{\widehat{\pi}} = Q^* - \widehat{Q}^{\pi^*} + \widehat{Q}^{\pi^*} - \widehat{Q}^* + \widehat{Q}^* - \widehat{Q}^{\widehat{\pi}} + \widehat{Q}^{\widehat{\pi}} - Q^{\widehat{\pi}}$$

$$\leq \left| Q^* - \widehat{Q}^{\pi^*} \right| + 0 + \epsilon_{\mathrm{PS}} + \left| \widehat{Q}^{\widehat{\pi}} - Q^{\widehat{\pi}} \right|.$$

which implies $\left\| Q^* - Q^{\widehat{\pi}} \right\|_\infty \leq \left\| Q^* - \widehat{Q}^{\pi^*} \right\|_\infty + \left\| \widehat{Q}^{\widehat{\pi}} - Q^{\widehat{\pi}} \right\|_\infty + \epsilon_{\mathrm{PS}}.$ ☐

**Lemma 2.** *We have*

$$Q^* - \widehat{Q}^{\pi^*} = (I - \gamma P^{\pi^*})^{-1}(\widehat{P} - P)\widehat{V}^{\pi^*},$$
$$Q^{\widehat{\pi}} - \widehat{Q}^{\widehat{\pi}} = (I - \gamma P^{\widehat{\pi}})^{-1}(\widehat{P} - P)\widehat{V}^{\widehat{\pi}}.$$

*Proof.* For any policy $\pi$,

$$Q^\pi - \widehat{Q}^\pi = (I - \gamma P^\pi)^{-1} r - (I - \gamma \widehat{P}^\pi)^{-1} r$$

$$= (I - \gamma P^\pi)^{-1}((I - \gamma \widehat{P}^\pi) - (I - \gamma P^\pi))\widehat{Q}^\pi$$

$$= \gamma (I - \gamma P^\pi)^{-1}(P^\pi - \widehat{P}^\pi)\widehat{Q}^\pi$$

$$= \gamma (I - \gamma P^\pi)^{-1}(P - \widehat{P})\widehat{V}^\pi.$$

Set $\pi = \pi^*$ and $\pi = \widehat{\pi}$, then we can get the results. ☐

**Lemma 3.** *For any value function $V$ and state action pair $(s, a)$,*

$$\phi(s,a)\sqrt{Var_{\mathcal{K}}(V)} = \sum_{k=1}^{d} \phi_k(s,a)\sqrt{Var_{s_k,a_k}(V)} \le \sqrt{Var_{s,a}(V)}.$$

*Proof.* Since $\phi(s,a)$ is a probability transition matrix, we can use Jensen's inequality here.

$$\phi(s,a)\sqrt{Var_{\mathcal{K}}(V)} \le \sqrt{\phi(s,a)Var_{\mathcal{K}}(V)}$$

$$= \sqrt{\sum_{k\in\mathcal{K}} \phi_k(s,a)Var_{s_k,a_k}(V)}$$

$$= \sqrt{\sum_{k\in\mathcal{K}}^{d} \phi_k(s,a)(P(s_k,a_k)V^2 - (P(s_k,a_k)V)^2)}$$

$$= \sqrt{\sum_{k\in\mathcal{K}}^{d} \phi_k(s,a)P(s_k,a_k)V^2 - \sum_{k\in\mathcal{K}}^{d} \phi_k(s,a)(P(s_k,a_k)V)^2}$$

$$\le \sqrt{P(s,a)V^2 - \Big(\sum_{k\in\mathcal{K}}^{d} \phi_k(s,a)P(s_k,a_k)V\Big)^2}$$

$$= \sqrt{P(s,a)V^2 - (P(s,a)V)^2}$$

$$= \sqrt{Var_{s,a}(V)}.$$

The two inequalities are due to Jensen's inequality and other steps are from $P(s,a) = \sum_{k\in\mathcal{K}} \phi_k(s,a)P(s_k,a_k)$, which is a row vector version of $P = \Phi P_{\mathcal{K}}$. $\square$

**Lemma 4.** *For any policy $\pi$ and $V^\pi$ is the value function in a MDP with transition $P$,*

$$\left\| (I - \gamma P^\pi)^{-1}\sqrt{Var_P(V^\pi)} \right\|_\infty \le \sqrt{\frac{2}{(1-\gamma)^3}}.$$

*Proof.* Since $(1-\gamma)(I - \gamma P^\pi)^{-1}$ is a probability transition matrix, we can apply Jensen's inequality,

$$\left\| (I - \gamma P^\pi)^{-1}\sqrt{Var_P(V^\pi)} \right\|_\infty \le \sqrt{\frac{1}{1-\gamma}}\sqrt{\|(I - \gamma P^\pi)^{-1}Var_P(V^\pi)\|_\infty}$$

$$\le \sqrt{\frac{2}{1-\gamma}}\sqrt{\|(I - \gamma^2 P^\pi)^{-1}Var_P(V^\pi)\|_\infty}$$

$$\le \sqrt{\frac{2}{1-\gamma}}\sqrt{\Sigma^\pi \cdot \frac{1}{\gamma^2}}$$

$$\le \sqrt{\frac{2}{(1-\gamma)^3}}$$

The definition of $\Sigma$ and a detailed proof is given in [6]. $\square$

**Lemma 5.** *For any value function $V_1$ and $V_2$, we have*

$$\sqrt{Var_{s,a}(V_1 + V_2)} \le \sqrt{Var_{s,a}(V_1)} + \sqrt{Var_{s,a}(V_2)}.$$

*Proof.* This Lemma is the triangle inequality for variance. $\square$

**Lemma 6.** *Let $u^\pi = \gamma(\widehat{P}(s,a) - P(s,a))\widehat{V}^\pi$ and $u^* = \gamma(\widehat{P}(s,a) - P(s,a))\widehat{V}^*$, then we have*

$$\widehat{Q}^\pi = \widetilde{Q}^\pi_{u^\pi}, \ \widehat{Q}^* = \widetilde{Q}^{\widehat{\pi}^*}_{u^*} = \widetilde{Q}^*_{u^*}, \ -\frac{1}{1-\gamma} \leq u^\pi \leq \frac{1}{1-\gamma}, \ -\frac{1}{1-\gamma} \leq u^* \leq \frac{1}{1-\gamma}.$$

*Proof.* Using the Bellman equation $Q^\pi = (I - \gamma P^\pi)^{-1}r$, we have

$$\begin{aligned}
\widetilde{Q}^\pi_{u^\pi} &= (I - \gamma \widetilde{P}^\pi)^{-1}(r + \Phi^{s,a} \cdot \gamma(\widehat{P}(s,a) - P(s,a))\widehat{V}^\pi) \\
&= (I - \gamma \widetilde{P}^\pi)^{-1}((I - \gamma \widehat{P}^\pi)\widehat{Q}^\pi + \gamma\Phi(\widehat{P}_\mathcal{K} - \widetilde{P}_\mathcal{K})\widehat{V}^\pi) \\
&= (I - \gamma \widetilde{P}^\pi)^{-1}((I - \gamma \widehat{P}^\pi)\widehat{Q}^\pi + \gamma(\widehat{P} - \widetilde{P})\widehat{V}^\pi) \\
&= (I - \gamma \widetilde{P}^\pi)^{-1}(I - \gamma \widetilde{P}^\pi)\widehat{Q}^\pi \\
&= \widehat{Q}^\pi.
\end{aligned}$$

Similarly, we have $\widehat{Q}^* = \widetilde{Q}^{\widehat{\pi}^*}_{u^*}$. By the definition of $\widehat{Q}^*$ and the sufficient condition for optimal value, we have

$$\widetilde{Q}^{\widehat{\pi}^*}_{u^*}(s, \widehat{\pi}^*(s)) = \widehat{Q}^*(s, \widehat{\pi}(s)) = \max_a \widehat{Q}^*(s,a) = \max_a \widetilde{Q}^{\widehat{\pi}^*}_{u^*}(s,a).$$

So, $\widehat{\pi}^*$ is the optimal policy in $\widetilde{\mathcal{M}}_{s,a,u^*}$ and $\widehat{Q}^* = \widetilde{Q}^{\widehat{\pi}^*}_{u^*} = \widetilde{Q}^*_{u^*}$.

As $0 \leq \widehat{V}^\pi \leq \frac{1}{1-\gamma}$ and $0 \leq \widehat{V}^* \leq \frac{1}{1-\gamma}$, we can immediately derive that $-\frac{1}{1-\gamma} \leq u^\pi \leq \frac{1}{1-\gamma}$, $-\frac{1}{1-\gamma} \leq u^* \leq \frac{1}{1-\gamma}$. □

**Remark 4.** *Lemma 6 shows that we can tune a bounded scalar $u$ in $\widetilde{M}_{s,a,u}$ to recover $\widehat{Q}$, which implies $\widehat{Q}$, as a function of $\widehat{P}(s,a)$, lies in a one-dimensional manifold in $\mathbb{R}^{|\mathcal{S}||\mathcal{A}|}$.*

**Lemma 7.** *For all $u_1, u_2 \in \mathbb{R}$ and policy $\pi$,*

$$\left\|\widetilde{Q}^\pi_{u_1} - \widetilde{Q}^\pi_{u_2}\right\|_\infty \leq \frac{1}{1-\gamma}|u_1 - u_2|, \left\|\widetilde{Q}^*_{u_1} - \widetilde{Q}^*_{u_2}\right\|_\infty \leq \frac{1}{1-\gamma}|u_1 - u_2|.$$

*Proof.* Using Bellman equation, we have

$$\begin{aligned}
\left\|\widetilde{Q}^\pi_{u_1} - \widetilde{Q}^\pi_{u_2}\right\|_\infty &= \left\|\left(I - \gamma\widetilde{P}^\pi\right)^{-1}(r + u_1\Phi^{s,a}) - \left(I - \gamma\widetilde{P}^\pi\right)^{-1}(r + u_2\Phi^{s,a})\right\|_\infty \\
&= \left\|(u_1 - u_2)\left(I - \gamma\widetilde{P}^\pi\right)^{-1}\Phi^{s,a}\right\|_\infty \\
&\leq |u_1 - u_2|\frac{1}{1-\gamma}\|\Phi^{s,a}\|_\infty \\
&\leq |u_1 - u_2|\frac{1}{1-\gamma}.
\end{aligned}$$

Now we prove the second claim. Set $\pi = \pi^*_{u_1}$ and $\pi = \pi^*_{u_2}$, we have

$$\left\|\widetilde{Q}^*_{u_1} - \widetilde{Q}^{\pi^*_{u_1}}_{u_2}\right\|_\infty \leq |u_1 - u_2|\frac{1}{1-\gamma}, \left\|\widetilde{Q}^{\pi^*_{u_2}}_{u_1} - \widetilde{Q}^*_{u_2}\right\|_\infty \leq |u_1 - u_2|\frac{1}{1-\gamma}.$$

Using the property of optimal value, we have

$$-|u_1 - u_2|\frac{1}{1-\gamma}\mathbf{1} \leq \widetilde{Q}^{\pi^*_{u_2}}_{u_1} - \widetilde{Q}^*_{u_2} \leq \widetilde{Q}^*_{u_1} - \widetilde{Q}^*_{u_2} \leq \widetilde{Q}^*_{u_1} - \widetilde{Q}^{\pi^*_{u_1}}_{u_2} \leq |u_1 - u_2|\frac{1}{1-\gamma}\mathbf{1},$$

which implies $\left\|\widetilde{Q}^*_{u_1} - \widetilde{Q}^*_{u_2}\right\|_\infty \leq |u_1 - u_2|\frac{1}{1-\gamma}$. □

**Remark 5.** *Lemma 7 shows that $\widehat{Q}_u$ is robust to $u$. This property implies that an $\epsilon$-net on $u$ can form an $\epsilon/(1-\gamma)$-net on $\widehat{Q}_u$.*

**Lemma 8.** *For a given finite set $B_{s,a}^\pi \subset U_{s,a}^\pi \cap [-\frac{1}{1-\gamma}, \frac{1}{1-\gamma}]$ and $\delta \geq 0$, with probability greater than $1 - \delta$, it holds for all $u \in B_{s,a}^\pi$ that*

$$\left| \left( P(s,a) - \widehat{P}(s,a) \right) \widetilde{V}_u^\pi \right| \leq \sqrt{\frac{2 \log \left( 4 \left| B_{s,a}^\pi \right| /\delta \right)}{N}} \sqrt{Var_{s,a}(\widetilde{V}_u^\pi)} + \frac{2 \log \left( 4 \left| B_{s,a}^\pi \right| /\delta \right)}{(1-\gamma)3N}.$$

*Similarly, For a given finite set $B_{s,a}^* \subset U_{s,a}^* \cap [-\frac{1}{1-\gamma}, \frac{1}{1-\gamma}]$ and $\delta \geq 0$, with probability greater than $1 - \delta$, it holds for all $u \in B_{s,a}^*$ that*

$$\left| \left( P(s,a) - \widehat{P}(s,a) \right) \widetilde{V}_u^* \right| \leq \sqrt{\frac{2 \log \left( 4 \left| B_{s,a}^* \right| /\delta \right)}{N}} \sqrt{Var_{s,a}(\widetilde{V}_u^*)} + \frac{2 \log \left( 4 \left| B_{s,a}^* \right| /\delta \right)}{(1-\gamma)3N}.$$

*Proof.* This is the direct application of Beinstein's inequality as $\widetilde{V}_u^\pi$ and $\widetilde{V}_u^*$ is independent of $\widehat{P}(s,a)$. $\qquad\square$

**Lemma 9.** *For a given finite set $B_{s,a}^{\pi^*} \subset U_{s,a}^{\pi^*} \cap [-\frac{1}{1-\gamma}, \frac{1}{1-\gamma}]$ and $B_{s,a}^* \subset U_{s,a}^* \cap [-\frac{1}{1-\gamma}, \frac{1}{1-\gamma}]$ and $\delta \geq 0$, with probability greater than $1 - 2\delta$, it holds for all $u \in B_{s,a}^{\pi^*}$ that*

$$\left| \left( P(s,a) - \widehat{P}(s,a) \right) \widehat{V}^{\pi^*} \right| \leq \sqrt{\frac{2 \log \left( 4 \left| B_{s,a}^{\pi^*} \right| /\delta \right)}{N}} \sqrt{Var_{s,a}(\widehat{V}^{\pi^*})} + \frac{2 \log \left( 4 \left| B_{s,a}^{\pi^*} \right| /\delta \right)}{(1-\gamma)3N}$$

$$+ \min_{u \in B_{s,a}^{\pi^*}} \left| u^{\pi^*} - u \right| \frac{1}{1-\gamma} \left( 1 + \sqrt{\frac{2 \log \left( 4 \left| B_{s,a}^{\pi^*} \right| /\delta \right)}{N}} \right),$$

$$\left| \left( P(s,a) - \widehat{P}(s,a) \right) \widehat{V}^* \right| \leq \sqrt{\frac{2 \log \left( 4 \left| B_{s,a}^* \right| /\delta \right)}{N}} \sqrt{Var_{s,a}(\widehat{V}^*)} + \frac{2 \log \left( 4 \left| B_{s,a}^* \right| /\delta \right)}{(1-\gamma)3N}$$

$$+ \min_{u \in B_{s,a}^*} \left| u^* - u \right| \frac{1}{1-\gamma} \left( 1 + \sqrt{\frac{2 \log \left( 4 \left| B_{s,a}^* \right| /\delta \right)}{N}} \right).$$

*Proof.* For the first claim, we have

$$\left| (P(s,a) - \widehat{P}(s,a))\widehat{V}^{\pi^*} \right|$$

$$\overset{(a)}{\leq} \left| (P(s,a) - \widehat{P}(s,a))\widetilde{V}_u^{\pi^*} \right| + \left| (P(s,a) - \widehat{P}(s,a))(\widehat{V}^{\pi^*} - \widetilde{V}_u^{\pi^*}) \right|$$

$$\overset{(b)}{\leq} \sqrt{\frac{2 \log(4 \left| B_{s,a}^{\pi^*} \right| /\delta)}{N}} \sqrt{Var_{s,a}(\widetilde{V}_u^{\pi^*})} + \frac{2 \log(4 \left| B_{s,a}^{\pi^*} \right| /\delta)}{(1-\gamma)3N} + \left\| \widehat{V}^{\pi^*} - \widetilde{V}_u^{\pi^*} \right\|_\infty$$

$$\overset{(c)}{\leq} \sqrt{\frac{2 \log(4 \left| B_{s,a}^{\pi^*} \right| /\delta)}{N}} \left( \sqrt{Var_{s,a}(\widehat{V}^{\pi^*})} + \sqrt{Var_{s,a}(\widetilde{V}_u^{\pi^*} - \widehat{V}^{\pi^*})} \right)$$

$$+ \frac{2 \log(4 \left| B_{s,a}^{\pi^*} \right| /\delta)}{(1-\gamma)3N} + \left\| \widehat{V}^{\pi^*} - \widetilde{V}_u^{\pi^*} \right\|_\infty$$

$$\overset{(d)}{\leq} \sqrt{\frac{2 \log(4 \left| B_{s,a}^{\pi^*} \right| /\delta)}{N}} \sqrt{Var_{s,a}(\widehat{V}^{\pi^*})} + \frac{2 \log(4 \left| B_{s,a}^{\pi^*} \right| /\delta)}{(1-\gamma)3N}$$

$$+ \left\| \widehat{V}^{\pi^*} - \widetilde{V}_u^{\pi^*} \right\|_\infty \left( 1 + \sqrt{\frac{2 \log(4 \left| B_{s,a}^\pi \right| /\delta)}{N}} \right)$$

$$\stackrel{(e)}{\leq} \sqrt{\frac{2\log(4\left|B^{\pi^*}_{s,a}\right|/\delta)}{N}}\sqrt{Var_{s,a}(\widehat{V}^{\pi^*})} + \frac{2\log(4\left|B^{\pi^*}_{s,a}\right|/\delta)}{(1-\gamma)3N}$$

$$+\left|u^{\pi^*}-u\right|\frac{1}{1-\gamma}\left(1+\sqrt{\frac{2\log(4\left|B^{\pi^*}_{s,a}\right|/\delta)}{N}}\right).$$

(a) is due to triangle inequality, (b) is from Lemma 8, (c) is from Lemma 5, (d) is due to the fact that $\sqrt{Var(V)}\leq V$ and (e) is from Lemma 7. As this equality holds for all $u\in B^{\pi^*}_{s,a}$, we can take minimum in the RHS, which proves the first claim. The second claims can proved in the same manner. □

**Lemma 10.** *For any given $\epsilon$ and all $(s,a)\in\mathcal{K}$, with probability larger than $1-\delta$,*

$$\left|\left(P(s,a)-\widehat{P}(s,a)\right)\widehat{V}^{\pi^*}\right| \leq \sqrt{\frac{2\log\left(32K/\delta\left(1-\gamma\right)^3\epsilon\right)}{N}}\sqrt{Var_{s,a}(\widehat{V}^{\pi^*})}$$

$$+\frac{2\log\left(32K/\delta\left(1-\gamma\right)^3\epsilon\right)}{(1-\gamma)\,3N} + \left(\sqrt{\frac{2\log\left(32K/\delta\left(1-\gamma\right)^3\epsilon\right)}{N}}+1\right)\frac{\epsilon\left(1-\gamma\right)}{4},$$

$$\left|\left(P(s,a)-\widehat{P}(s,a)\right)\widehat{V}^*\right| \leq \sqrt{\frac{2\log\left(32K/\delta\left(1-\gamma\right)^3\epsilon\right)}{N}}\sqrt{Var_{s,a}\left(\widehat{V}^*\right)}$$

$$+\frac{2\log\left(32K/\delta\left(1-\gamma\right)^3\epsilon\right)}{(1-\gamma)\,3N} + \left(\sqrt{\frac{2\log\left(32K/\delta\left(1-\gamma\right)^3\epsilon\right)}{N}}+1\right)\frac{\epsilon\left(1-\gamma\right)}{4}.$$

*For simplicity, we set $c(\delta,\gamma,\epsilon)=2\log\left(32K/\delta\left(1-\gamma\right)^3\epsilon\right)$ and we use $c$ to represent $c(\delta,\gamma,\epsilon)$ as it includes only log factors.*

*Proof.* We set $B^{\pi^*}_{s,a}$ to be the evenly spaced elements in the interval $U^*_{s,a}\cap[-\frac{1}{1-\gamma},\frac{1}{1-\gamma}]$ and $\left|B^{\pi^*}_{s,a}\right|=\frac{4}{(1-\gamma)^3\epsilon}$. Then for any $u'\in U^*_{s,a}\cap[-\frac{1}{1-\gamma},\frac{1}{1-\gamma}]$, we have $\min_{u\in B^{\pi^*}_{s,a}}|u'-u|\leq(1-\gamma)^2\epsilon/4$. Note that $u^{\pi^*}\in U^*_{s,a}\cap[-\frac{1}{1-\gamma},\frac{1}{1-\gamma}]$. Then Lemma 9 implies this result. Similarly we can prove the second claim. □

**Lemma 11.** *With probability larger than $1-\delta$,*

$$\left|(P-\widehat{P})\widehat{V}^{\pi^*}\right| \leq \sqrt{\frac{c}{N}}\sqrt{Var_P(\widehat{V}^{\pi^*})} + \left[\frac{c}{(1-\gamma)\,3N}+\left(\sqrt{\frac{c}{N}}+1\right)\frac{\epsilon\left(1-\gamma\right)}{4}\right]\mathbf{1},$$

$$\left|(P-\widehat{P})\widehat{V}^*\right| \leq \sqrt{\frac{c}{N}}\sqrt{Var_P(\widehat{V}^*)} + \left[\frac{c}{(1-\gamma)\,3N}+\left(\sqrt{\frac{c}{N}}+1\right)\frac{\epsilon(1-\gamma)}{4}\right]\mathbf{1}.$$

*Proof.*

$$\left|(P-\widehat{P})\widehat{V}^{\pi^*}\right| = \left|\Phi(P_\mathcal{K}-\widehat{P}_\mathcal{K})\widehat{V}^*\right|$$

$$\stackrel{(a)}{\leq} \Phi\left|(P_\mathcal{K}-\widehat{P}_\mathcal{K})\widehat{V}^*\right|$$

$$\stackrel{(b)}{\leq} \Phi\sqrt{\frac{c}{N}}\sqrt{Var_\mathcal{K}(\widehat{V}^{\pi^*})} + \Phi\left[\frac{c}{(1-\gamma)\,3N}+\left(\sqrt{\frac{c}{N}}+1\right)\frac{\epsilon\left(1-\gamma\right)}{4}\right]\mathbf{1}$$

$$\overset{(c)}{\leq} \sqrt{\frac{c}{N}} \sqrt{Var_P \left( \widehat{V}^{\pi^*} \right)} + \left[ \frac{c}{(1-\gamma)\,3N} + \left( \sqrt{\frac{c}{N}} + 1 \right) \frac{\epsilon\,(1-\gamma)}{4} \right] \mathbf{1}.$$

(a) is due to $\Phi$ is non-negative, (b) is from Lemma 10 and (c) is from Lemma 3. The second claim can be proved in the same manner. $\qquad\square$

**Lemma 12.** *With probability larger than* $1-\delta$*, and* $\widehat{\pi}$ *is a* $\epsilon_{\mathrm{PS}}$*-optimal policy in* $\widehat{M}$*.*

$$\left\| Q^* - \widehat{Q}^{\pi^*} \right\|_\infty \leq \frac{\gamma}{1-\alpha} \left( \sqrt{\frac{c}{N\,(1-\gamma)^3}} + \frac{c}{(1-\gamma)^2\,3N} + \left( \sqrt{\frac{c}{N}} + 1 \right) \frac{\epsilon}{4} \right),$$

$$\left\| Q^{\widehat{\pi}} - \widehat{Q}^{\widehat{\pi}} \right\|_\infty \leq \frac{\gamma}{1-\alpha} \left( \sqrt{\frac{c}{N\,(1-\gamma)^3}} + \frac{c}{(1-\gamma)^2\,3N} + \left( \sqrt{\frac{c}{N}} + 1 \right) \left( \frac{\epsilon}{4} + \frac{\epsilon_{\mathrm{PS}}}{1-\gamma} \right) \right).$$

*where* $\alpha = \alpha(\delta, \gamma, \epsilon, N) = \sqrt{\frac{c(\delta,\gamma,\epsilon)}{N(1-\gamma)^2}}.$

*Proof.*

$$\left\| Q^* - \widehat{Q}^{\pi^*} \right\|_\infty$$

$$= \left\| \left( I - \gamma P^{\pi^*} \right)^{-1} \left( \widehat{P} - P \right) \widehat{V}^{\pi^*} \right\|_\infty$$

$$\overset{(a)}{\leq} \left\| \left( I - \gamma P^{\pi^*} \right)^{-1} \left[ \sqrt{\frac{c}{N}} \sqrt{Var_P \left( \widehat{V}^{\pi^*} \right)} + \frac{c}{(1-\gamma)\,3N} + \left( \sqrt{\frac{c}{N}} + 1 \right) \frac{\epsilon}{4\,(1-\gamma)} \right] \right\|_\infty$$

$$\overset{(b)}{\leq} \sqrt{\frac{c}{N\,(1-\gamma)^3}} + \sqrt{\frac{c}{N}} \frac{\left\| Q^* - \widehat{Q}^{\pi^*} \right\|}{1-\gamma} + \frac{c}{(1-\gamma)^2\,3N} + \left( \sqrt{\frac{c}{N}} + 1 \right) \frac{\epsilon}{4}.$$

(a) is from Lemma 11 and (b) is from Lemma 4. Solving for $\left\| Q^* - \widehat{Q}^{\pi^*} \right\|_\infty$ proves the first claim.

$$\left\| Q^{\widehat{\pi}} - \widehat{Q}^{\widehat{\pi}} \right\|_\infty$$

$$= \left\| \left( I - \gamma P^{\widehat{\pi}} \right)^{-1} \left( \widehat{P} - P \right) \widehat{V}^{\widehat{\pi}} \right\|_\infty$$

$$\overset{(a)}{\leq} \left\| \left( I - \gamma P^{\widehat{\pi}} \right)^{-1} \left( \widehat{P} - P \right) \widehat{V}^* \right\|_\infty + \left\| \left( I - \gamma P^{\widehat{\pi}} \right)^{-1} \left( \widehat{P} - P \right) \left( \widehat{V}^{\widehat{\pi}} - \widehat{V}^* \right) \right\|_\infty$$

$$\overset{(b)}{\leq} \left\| \left( I - \gamma P^{\widehat{\pi}} \right)^{-1} \left[ \sqrt{\frac{c}{N}} \sqrt{Var_P(\widehat{V}^*)} + \frac{c}{(1-\gamma)3N} + \left( \sqrt{\frac{c}{N}} + 1 \right) \frac{\epsilon}{4(1-\gamma)} \right] \right\|_\infty$$
$$+ \frac{\epsilon_{\mathrm{PS}}}{1-\gamma}$$

$$\overset{(c)}{\leq} \left\| \left( I - \gamma P^{\widehat{\pi}} \right)^{-1} \left[ \sqrt{\frac{c}{N}} \sqrt{Var_P \left( V^{\widehat{\pi}} + \widehat{V}^{\widehat{\pi}} - V^{\widehat{\pi}} + \widehat{V}^* - \widehat{V}^{\widehat{\pi}} \right)} + \frac{c}{(1-\gamma)3N} \right. \right.$$
$$\left. \left. + \left( \sqrt{\frac{c}{N}} + 1 \right) \frac{\epsilon}{4(1-\gamma)} \right] \right\|_\infty + \frac{\epsilon_{\mathrm{PS}}}{1-\gamma}$$

$$\overset{(d)}{\leq} \left\| \sqrt{\frac{c}{N}} \left( I - \gamma P^{\widehat{\pi}} \right)^{-1} \sqrt{Var_P \left( \widehat{V}^{\widehat{\pi}} \right)} \right\|_\infty + \sqrt{\frac{c}{N}} \frac{\left\| Q^{\widehat{\pi}} - \widehat{Q}^{\widehat{\pi}} \right\|_\infty}{1-\gamma} + \frac{c}{(1-\gamma)^2 3N}$$
$$+ \left( \sqrt{\frac{c}{N}} + 1 \right) \left( \frac{\epsilon}{4} + \frac{\epsilon_{\mathrm{PS}}}{1-\gamma} \right)$$

$$\overset{(e)}{\leq} \sqrt{\frac{c}{N(1-\gamma)^3}} + \sqrt{\frac{c}{N}} \frac{\left\| Q^{\widehat{\pi}} - \widehat{Q}^{\widehat{\pi}} \right\|_\infty}{1-\gamma} + \frac{c}{(1-\gamma)^2 3N} + \left( \sqrt{\frac{c}{N}} + 1 \right) \left( \frac{\epsilon}{4} + \frac{\epsilon_{\mathrm{PS}}}{1-\gamma} \right).$$

(a), (c), (d) are due to triangle inequality, (b) is from Lemma 11 and (e) is from Lemma 4. Solving for $\left\| Q^{\widehat{\pi}} - \widehat{Q}^{\widehat{\pi}} \right\|_\infty$ proves the second claim. $\qquad\square$

*Proof of Theorem 1.* From Lemma 1, with probability larger than $1 - \delta$, we have

$$
\left\|Q^* - Q^{\widehat{\pi}}\right\|_\infty \overset{(a)}{\le} \left\|Q^* - \widehat{Q}^{\pi^*}\right\|_\infty + \left\|\widehat{Q}^{\widehat{\pi}} - Q^{\widehat{\pi}}\right\|_\infty + \epsilon_{\mathrm{PS}}
$$

$$
\overset{(b)}{\le} \frac{\gamma}{1-\alpha}\left[2\left(\sqrt{\frac{c}{N(1-\gamma)^3}} + \frac{c}{(1-\gamma)^2 3N} + \left(\sqrt{\frac{c}{N}}+1\right)\frac{\epsilon}{4}\right)\right.
$$

$$
\left. + \left(\sqrt{\frac{c}{N}}+1\right)\frac{\epsilon_{\mathrm{PS}}}{1-\gamma}\right] + \epsilon_{\mathrm{PS}}
$$

(a) is from Lemma 1 and (b) is from Lemma 12. For $N \ge \frac{C\log(CK(1-\gamma)^{-1}\delta^{-1}\epsilon^{-1})}{(1-\gamma)^3\epsilon^2}$ with proper constant $C$, we have $\frac{\gamma}{1-\alpha(\delta,\gamma,\epsilon,N)}\left(\sqrt{\frac{c(\delta,\gamma,\epsilon)}{N}}+1\right) \le 2$, thus

$$
\left\|V^* - V^{\widehat{\pi}}\right\|_\infty \le \epsilon + \frac{3\epsilon_{\mathrm{PS}}}{1-\gamma},
$$

which completes the proof.

$\square$

Now we prove Theorem 2, where the transition model $P$ can be approximated by linear transition model, i.e. $P = \bar{P} + \Xi = \Phi\bar{P}_{\mathcal{K}} + \Xi$, where $\bar{P}$ is a linear transition model and $\Xi$ is the approximation error matrix. We set $\xi = \|\Xi\|_{1,\infty}$.

**Lemma 13.** *For any value function $V$ and state action pair $(s, a)$,*

$$
\phi(s,a)\sqrt{Var_{\mathcal{K}}(V)} = \sum_{k\in\mathcal{K}}\phi_k(s,a)\sqrt{Var_{s_k,a_k}(V)} \le \sqrt{Var_{s,a}(V)} + 2\sqrt{\frac{3\xi}{(1-\gamma)^2}}.
$$

*Proof.* Since $\phi(s,a)$ is a probability transition matrix, we can use Jensen's inequality here.

$$
\phi(s,a)\sqrt{Var_{\mathcal{K}}(V)}
$$

$$
\le \sqrt{\phi(s,a)Var_{\mathcal{K}}(V)}
$$

$$
= \sqrt{\sum_{k\in\mathcal{K}}\phi_k(s,a)Var_{s_k,a_k}(V)}
$$

$$
= \sqrt{\sum_{k\in\mathcal{K}}\phi_k(s,a)\left(P(s_k,a_k)V^2 - (P(s_k,a_k)V)^2\right)}
$$

$$
= \sqrt{\sum_{k\in\mathcal{K}}\phi_k(s,a)P(s_k,a_k)V^2 - \sum_{k\in\mathcal{K}}\phi_k(s,a)\left(P(s_k,a_k)V\right)^2}
$$

$$
= \sqrt{\sum_{k\in\mathcal{K}}\phi_k(s,a)\left(\bar{P}(s_k,a_k)V^2 + \Xi(s_k,a_k)V^2\right) - \sum_{k\in\mathcal{K}}\phi_k(s,a)\left(\left(\bar{P}(s_k,a_k) + \Xi(s_k,a_k)\right)V\right)^2}
$$

$$
\le \sqrt{\sum_{k\in\mathcal{K}}\phi_k(s,a)\bar{P}(s_k,a_k)V^2 - \sum_{k\in\mathcal{K}}\phi_k(s,a)\left(\bar{P}(s_k,a_k)V\right)^2} + \sqrt{\frac{3\xi}{(1-\gamma)^2}}
$$

$$
\le \sqrt{\bar{P}(s,a)V^2 - \sum_{k\in\mathcal{K}}\phi_k(s,a)\left(\bar{P}(s_k,a_k)V\right)^2} + \sqrt{\frac{3\xi}{(1-\gamma)^2}}
$$

$$
= \sqrt{\bar{P}(s,a)V^2 - \left(\bar{P}(s,a)V\right)^2} + \sqrt{\frac{3\xi}{(1-\gamma)^2}}
$$

$$\leq \sqrt{P(s,a)V^2 - (P(s,a)V)^2} + 2\sqrt{\frac{3\xi}{(1-\gamma)^2}}$$

$$= \sqrt{Var_{s,a}(V)} + 2\sqrt{\frac{3\xi}{(1-\gamma)^2}}.$$

$\square$

**Lemma 14.** *With probability larger than $1 - \delta$, and $\widehat{\pi}$ is a $\epsilon_{\mathrm{PS}}$-optimal policy in $\widehat{\mathcal{K}}$.*

$$\left\| Q^* - \widehat{Q}^{\pi^*} \right\|_\infty \leq \frac{\gamma}{1-\alpha}\left( \sqrt{\frac{c}{N(1-\gamma)^3}} + \frac{c}{(1-\gamma)^2 3N} + \left(\sqrt{\frac{c}{N}} + 1\right)\frac{\epsilon}{4} + 8\sqrt{\frac{\xi}{(1-\gamma)^4}} \right),$$

$$\left\| Q^{\widehat{\pi}} - \widehat{Q}^{\widehat{\pi}} \right\|_\infty \leq \frac{\gamma}{1-\alpha}\left( \sqrt{\frac{c}{N(1-\gamma)^3}} + \frac{c}{(1-\gamma)^2 3N} + \left(\sqrt{\frac{c}{N}} + 1\right)\left(\frac{\epsilon}{4} + \frac{\epsilon_{\mathrm{PS}}}{1-\gamma}\right)\right.$$
$$\left. + 8\sqrt{\frac{\xi}{(1-\gamma)^4}} \right).$$

*Proof.*

$$\left\| Q^* - \widehat{Q}^{\pi^*} \right\|_\infty$$
$$= \left\| \left(I - \gamma P^{\pi^*}\right)^{-1}(\widehat{P} - P)\widehat{V}^{\pi^*} \right\|_\infty$$
$$= \left\| \left(I - \gamma P^{\pi^*}\right)^{-1}\left(\Phi\widehat{P}_\mathcal{K} - \Phi\bar{P}_\mathcal{K} - \Xi_\mathcal{K}\right)\widehat{V}^{\pi^*} \right\|_\infty$$
$$\leq \left\| \left(I - \gamma P^{\pi^*}\right)^{-1}\Phi\left(\widehat{P}_\mathcal{K} - \bar{P}_\mathcal{K} - \Xi_\mathcal{K}\right)\widehat{V}^{\pi^*} \right\|_\infty + \frac{2\xi}{(1-\gamma)^2}$$
$$= \left\| \left(I - \gamma P^{\pi^*}\right)^{-1}\Phi\left(\widehat{P}_\mathcal{K} - P_\mathcal{K}\right)\widehat{V}^{\pi^*} \right\|_\infty + \frac{2\xi}{(1-\gamma)^2}$$
$$\leq \left\| \left(I - \gamma P^{\pi^*}\right)^{-1}\left[\sqrt{\frac{c}{N}}\left(\sqrt{Var_P\left(\widehat{V}^{\pi^*}\right)} + 2\sqrt{\frac{3\xi}{(1-\gamma)^2}}\right) + \frac{c}{(1-\gamma)3N}\right.\right.$$
$$\left.\left. + \left(\sqrt{\frac{c}{N}} + 1\right)\frac{\epsilon}{4(1-\gamma)}\right] \right\|_\infty + \frac{2\xi}{(1-\gamma)^2}$$
$$\leq \sqrt{\frac{c}{N(1-\gamma)^3}} + \sqrt{\frac{c}{N}}\frac{\left\| Q^* - \widehat{Q}^{\pi^*} \right\|}{1-\gamma} + \frac{c}{(1-\gamma)^2 3N} + \left(\sqrt{\frac{c}{N}} + 1\right)\frac{\epsilon}{4} + 8\sqrt{\frac{\xi}{(1-\gamma)^4}}.$$

Solving for $\left\| Q^* - \widehat{Q}^{\pi^*} \right\|_\infty$ proves the first claim. The second claim can be proved in a similar manner. $\square$

*Proof of Theorem 2.* From Lemma 1, with probability larger than $1 - \delta$, we have

$$\left\| Q^* - Q^{\widehat{\pi}} \right\|_\infty$$
$$\leq \left\| Q^* - \widehat{Q}^{\pi^*} \right\|_\infty + \left\| \widehat{Q}^{\widehat{\pi}} - Q^{\widehat{\pi}} \right\|_\infty + \epsilon_{\mathrm{PS}}$$
$$\leq \frac{\gamma}{1-\alpha}\left[2\left(\sqrt{\frac{c}{N(1-\gamma)^3}} + \frac{c}{(1-\gamma)^2 3N} + \left(\sqrt{\frac{c}{N}} + 1\right)\frac{\epsilon}{4} + 8\sqrt{\frac{\xi}{(1-\gamma)^4}}\right)\right.$$
$$\left. + \left(\sqrt{\frac{c}{N}} + 1\right)\frac{\epsilon_{\mathrm{PS}}}{1-\gamma}\right] + \epsilon_{\mathrm{PS}}$$

For $N \geq \frac{C \log(CK(1-\gamma)^{-1}\delta^{-1})\epsilon^{-1}}{(1-\gamma)^3 \epsilon^2}$ with proper constant $C$, we have $\frac{\gamma}{1-\alpha}\left(\sqrt{\frac{c}{N}}+1\right) \leq 2$, thus

$$\left\|V^* - V^{\widehat{\pi}}\right\|_\infty \leq \epsilon + \frac{3\epsilon_{\mathrm{PS}}}{1-\gamma} + \frac{16\sqrt{\xi}}{(1-\gamma)^2},$$

which completes the proof. $\qquad\square$

## C.2 Sample Complexity for Finite Horizon MDP

Here we prove the sample complexity result for FHMDP using the auxiliary MDP technique. The difference with DMDP is that here we need to tune the reward in each time step.

**Definition 2.** (Auxiliary Model) For an estimated transition model $\widehat{\mathcal{M}} = (\mathcal{S}, \mathcal{A}, \widehat{P} = \Phi\widehat{P}_{\mathcal{K}}, r, \gamma)$ and a given anchor state pair $(s, a)$, the auxiliary transition model is $\widetilde{\mathcal{M}}_{s,a,u} = (\mathcal{S}, \mathcal{A}, \widetilde{P} = \Phi\widetilde{P}_K, \widetilde{r}_h^u = r + u_h \Phi^{s,a}, \gamma)$, where

$$\widetilde{P}_{\mathcal{K}}(s', a') = \begin{cases} \widehat{P}(s', a') & \text{if } (s', a') \neq (s, a) \\ P(s, a) & \text{otherwise,} \end{cases}$$

$\Phi^{s,a}$ is the column of $\Phi$ that corresponds to anchor state $(s, a)$, $\widetilde{r}_h^u$ is the reward in step $h$ and $u = (u_0, u_1, \cdots, u_{H-1})$ is a $H$ dimensional vector that will be determined latter.

**Remark 6.** *The reward in $\widetilde{\mathcal{M}}_{s,a,u}$ may not be stationary, which means $\widetilde{r}_0^u, \widetilde{r}_1^u, \cdots, \widetilde{r}_{H-1}^u$ can be different.*

**Definition 3.** (Feasible Set for $u$) For the auxiliary transition model $\widetilde{\mathcal{M}}_{s,a,u}$, $U_{s,a}^\pi$ is defined as the set of $u$ such that $\tilde{V}_{h,u}^\pi \in [0, H-h]^{\mathcal{S}}, \forall h \in [H]$ and $U_{s,a}^*$ is defined as the set of $u$ such that $\tilde{V}_{h,u}^* \in [0, H-h]^{\mathcal{S}}, \forall h \in [H]$.

**Remark 7.** *$u$ that satisfies $0 \leq r + u_h\Phi^{s,a} \leq 1, \forall h \in [H]$ is in both $U_{s,a}^\pi$ and $U_{s,a}^*$ for arbitrary $\pi$. Immediately we have $\mathbf{0} \in U_{s,a}^\pi$ and $\mathbf{0} \in U_{s,a}^*$ for arbitrary $\pi$. Note that both $U_{s,a}^\pi$ and $U_{s,a}^*$ are independent of $\widehat{P}(s,a)$ and are intervals.*

**Notations** For simplicity, we ignore $(s, a)$ in functions of auxiliary transition model $\mathcal{M}_{s,a,u}$. We use $\widetilde{V}_{h,u}^\pi, \widetilde{Q}_{h,u}^\pi$ to denote value function and Q-function in step $h$ and $\widetilde{\pi}_u^*$ to be the optimal policy in $\mathcal{M}_{s,a,u}$.

**Lemma 15.** *For a $\epsilon_{\mathrm{PS}}$-optimal policy $\widehat{\pi} = (\widehat{\pi}_0, \cdots, \widehat{\pi}_{H-1})$ in $\widehat{\mathcal{M}}$, we have*

$$\left\|Q_0^* - Q_0^{\widehat{\pi}}\right\|_\infty \leq \left\|V_0^* - \widehat{V}_0^{\pi^*}\right\|_\infty + \left\|\widehat{V}_0^{\widehat{\pi}} - V_0^{\widehat{\pi}}\right\|_\infty + \epsilon_{\mathrm{PS}}.$$

*Proof.*

$$0 < Q_0^* - Q_0^{\widehat{\pi}} = Q_0^* - \widehat{Q}_0^{\pi^*} + \widehat{Q}_0^{\pi^*} - \widehat{Q}_0^* + \widehat{Q}_0^* - \widehat{Q}_0^{\widehat{\pi}} + \widehat{Q}_0^{\widehat{\pi}} - Q_0^{\widehat{\pi}}$$

$$\leq \left|Q_0^* - \widehat{Q}_0^{\pi^*}\right| + 0 + \epsilon_{\mathrm{PS}} + \left|\widehat{Q}_0^{\widehat{\pi}} - Q_0^{\widehat{\pi}}\right|,$$

which implies $\left\|Q_0^* - Q_0^{\widehat{\pi}}\right\|_\infty \leq \left\|Q_0^* - \widehat{Q}_0^{\pi^*}\right\|_\infty + \left\|\widehat{Q}_0^{\widehat{\pi}} - Q_0^{\widehat{\pi}}\right\|_\infty + \epsilon_{\mathrm{PS}}.$ $\qquad\square$

**Lemma 16.** *For FHMDP, we have*

$$Q_0^\pi - \widehat{Q}_0^\pi = \sum_{h=0}^{H-1} \prod_{i=0}^{h-1} P^{\pi_i}(P - \widehat{P})\widehat{V}_{h+1}^\pi.$$

*Proof.* Using Bellman equation, we have

$$
\begin{aligned}
Q_0^\pi - \widehat{Q}_0^\pi &= (r + PV_1^\pi) - (r + \widehat{P}\widetilde{V}_1^\pi) \\
&= P(V_1^\pi - \widehat{V}_1^\pi) + (P - \widehat{P})\widetilde{V}_1^\pi \\
&= P^{\pi_0}(Q_1^\pi - \widehat{Q}_1^\pi) + (P - \widehat{P})\widetilde{V}_1^\pi \\
&= P^{\pi_0} P^{\pi_1}(Q_2^\pi - \widehat{Q}_2^\pi) + P^{\pi_0}(P - \widehat{P})\widetilde{V}_2^\pi + (P - \widehat{P})\widetilde{V}_1^\pi \\
&= \sum_{h=0}^{H-1} \prod_{i=0}^{h-1} P^{\pi_i}(P - \widehat{P})\widehat{V}_{h+1}^\pi.
\end{aligned}
$$

The last equality is derived by iteratively expand $Q_h^\pi - \widehat{Q}_h^\pi$. $\qquad\square$

**Lemma 17.** *For any policy $\pi$ and $V^\pi$ is the value function in a MDP with transition $P$,*

$$
\left\| \sum_{h=0}^{H-1} \prod_{i=0}^{h-1} P^{\pi_i} \sqrt{Var_P(V_{h+1}^\pi)} \right\|_\infty \le \sqrt{2H^3}.
$$

*Proof.* The proof is similar to the case in DMDP and can be found in [3]. $\qquad\square$

**Lemma 18.** *Let $u_h^\pi = \left(\widehat{P}(s,a) - P(s,a)\right)\widehat{V}_{h+1}^\pi, \forall h \in [H]$ and $u_h^* = \left(\widehat{P}(s,a) - P(s,a)\right)\widehat{V}_{h+1}^*, \forall h \in [H]$, then we have*

$$
\widehat{Q}_h^\pi = \widetilde{Q}_{h,u^\pi}^{\widehat{\pi}}, \ \widehat{Q}_h^* = \widetilde{Q}_{h,u^*}^{\widehat{\pi}} = \widetilde{Q}_{h,u^*}^*, |u_h^\pi| \le H - h - 1, |u_h^*| \le H - h - 1, \forall h \in [H].
$$

*Proof.* We provethis argument by mathematical induction. When $h = H - 1$, we have $u_h^\pi = \left(\widehat{P}(s,a) - P(s,a)\right)\widehat{V}_H^\pi = 0$ and $\widehat{Q}_h^\pi = r = r + u_h^\pi \Phi^{s,a} = \widetilde{Q}_{h,u^\pi}^\pi$.

If the argument holds for $h + 1$, then we have

$$
\begin{aligned}
\widehat{Q}_h^\pi &= r + \widehat{P}\widehat{V}_{h+1}^\pi \\
&= r + \widehat{P}\widetilde{V}_{h+1}^\pi \\
&= r + \Phi\left(\widehat{P}_{\mathcal{K}} - \widetilde{P}_{\mathcal{K}}\right)\widehat{V}_{h+1}^\pi + \widetilde{P}\widetilde{V}_{h+1}^\pi \\
&= r + \left(\widehat{P}(s,a) - P(s,a)\right)\widehat{V}_{h+1}^\pi \Phi^{s,a} + \widetilde{P}\widetilde{V}_{h+1}^\pi \\
&= r + u_h^\pi \Phi^{s,a} + \widetilde{P}\widetilde{V}_{h+1}^\pi \\
&= \widetilde{Q}_{h,u^\pi}^\pi.
\end{aligned}
$$

As $\widehat{V}_{h+1}^\pi \in [0, H - h - 1]$, we have $u_h^\pi = (\widehat{P}(s,a) - P(s,a))\widehat{V}_{h+1}^\pi \in [h + 1 - H, H - h - 1]$. The proof for $\widehat{\pi}^*$ is identical. $\qquad\square$

**Lemma 19.** *For all $u, u' \in \mathbb{R}^H$ and policy $\pi$,*

$$
\left\|\widetilde{Q}_{h,u}^\pi - \widetilde{Q}_{h,u'}^\pi\right\|_\infty \le (H - h)\left\|u - u'\right\|_\infty, \left\|\widetilde{Q}_{h,u}^* - \widetilde{Q}_{h,u'}^*\right\|_\infty \le (H - h)\left\|u - u'\right\|_\infty.
$$

*Proof.* We prove the first claim:

$$
\begin{aligned}
\left\|\widetilde{Q}_{h,u}^\pi - \widetilde{Q}_{h,u'}^\pi\right\|_\infty &= \left\|\left(r + u_h \Phi^{s,a} + \widetilde{P}V_{h,u}^\pi\right) - \left(r + u_h' \Phi^{s,a} + \widetilde{P}V_{h,u'}^\pi\right)\right\|_\infty \\
&\le \left\|(u_h - u_h')\Phi^{s,a}\right\|_\infty + \left\|\widetilde{P}^\pi\left(\widetilde{Q}_{h+1,u}^\pi - \widetilde{Q}_{h+1,u'}^\pi\right)\right\|_\infty
\end{aligned}
$$

$$\leq |u_h - u'_h| + \left\| \widetilde{Q}^\pi_{h+1,u} - \widetilde{Q}^\pi_{h+1,u'} \right\|_\infty$$

$$\leq \sum_{i=h}^{H-1} |u_i - u'_i|$$

$$\leq (H-h) \left\| u - u' \right\|_\infty$$

The proof for the second claim is identical. □

**Lemma 20.** *For a given finite set $B^{\pi^*}_{s,a} \subset U^{\pi^*}_{s,a} \cap [-H,H]^H$ and $\delta \geq 0$, with probability greater than $1 - \delta$, it holds for all $u \in B^\pi_{s,a}$ that*

$$\left| \left( P(s,a) - \widehat{P}(s,a) \right) \widetilde{V}^{\pi^*}_{h,u} \right| \leq \sqrt{\frac{2 \log \left( 4 \left| B^{\pi^*}_{s,a} \right| / \delta \right)}{N}} \sqrt{Var_{s,a}(\widetilde{V}^{\pi^*}_{h,u})} + \frac{2 \log \left( 4 \left| B^{\pi^*}_{s,a} \right| / \delta \right) (H - h)}{3N}.$$

*Similarly, For a given finite set $B^*_{s,a} \subset U^*_{s,a} \cap [-H,H]^H$ and $\delta \geq 0$, with probability greater than $1 - \delta$, it holds for all $u \in B^*_{s,a}$ that*

$$\left| \left( P(s,a) - \widehat{P}(s,a) \right) \widetilde{V}^*_{h,u} \right| \leq \sqrt{\frac{2 \log \left( 4 \left| B^*_{s,a} \right| / \delta \right)}{N}} \sqrt{Var_{s,a}(\widetilde{V}^*_{h,u})} + \frac{2 \log \left( 4 \left| B^*_{s,a} \right| / \delta \right) (H - h)}{3N}.$$

*Proof.* This is the direct application of Beinstein's inequality as $\widetilde{V}^\pi_{h,u}$ and $\widetilde{V}^*_{h,u}$ is independent of $\widehat{P}(s,a)$. □

**Lemma 21.** *For a given finite set $B^{\pi^*}_{s,a} \subset U^{\pi^*}_{s,a} \cap [-H,H]^H$ and $B^*_{s,a} \subset U^*_{s,a} \cap [-H,H]^H$ and $\delta \geq 0$, with probability greater than $1 - 2H\delta$, it holds for all $u \in B^{\pi^*}_{s,a}$ and $h \in [H]$ that*

$$\left| \left( P(s,a) - \widehat{P}(s,a) \right) \widehat{V}^{\pi^*}_h \right| \leq \sqrt{\frac{2 \log \left( 4 \left| B^{\pi^*}_{s,a} \right| / \delta \right)}{N}} \sqrt{Var_{s,a}(\widehat{V}^{\pi^*}_h)} + \frac{2 \log \left( 4 \left| B^{\pi^*}_{s,a} \right| / \delta \right) (H - h)}{3N}$$

$$+ \min_{u \in B^{\pi^*}_{s,a}} \left\| u^{\pi^*} - u \right\|_\infty (H - h) \left( 1 + \sqrt{\frac{2 \log \left( 4 \left| B^{\pi^*}_{s,a} \right| / \delta \right)}{N}} \right),$$

$$\left| \left( P(s,a) - \widehat{P}(s,a) \right) \widehat{V}^*_h \right| \leq \sqrt{\frac{2 \log \left( 4 \left| B^*_{s,a} \right| / \delta \right)}{N}} \sqrt{Var_{s,a}(\widehat{V}^*_h)} + \frac{2 \log \left( 4 \left| B^*_{s,a} \right| / \delta \right) (H - h)}{3N}$$

$$+ \min_{u \in B^*_{s,a}} \left\| u^* - u \right\|_\infty (H - h) \left( 1 + \sqrt{\frac{2 \log \left( 4 \left| B^*_{s,a} \right| / \delta \right)}{N}} \right).$$

*Proof.* We have

$$\left| \left( P(s,a) - \widehat{P}(s,a) \right) \widehat{V}^{\pi^*}_h \right|$$

$$\overset{(a)}{\leq} \left| \left( P(s,a) - \widehat{P}(s,a) \right) \widetilde{V}^{\pi^*}_{h,u} \right| + \left| \left( P(s,a) - \widehat{P}(s,a) \right) \left( \widehat{V}^{\pi^*}_h - \widetilde{V}^{\pi^*}_{h,u} \right) \right|$$

$$\overset{(b)}{\leq} \sqrt{\frac{2 \log \left( 4 \left| B^{\pi^*}_{s,a} \right| / \delta \right)}{N}} \sqrt{Var_{s,a}(\widetilde{V}^{\pi^*}_{h,u})} + \frac{2 \log \left( 4 \left| B^{\pi^*}_{s,a} \right| / \delta \right) (H - h)}{3N} + \left\| \widehat{V}^{\pi^*}_h - \widetilde{V}^{\pi^*}_{h,u} \right\|_\infty$$

$$\overset{(c)}{\leq} \sqrt{\frac{2 \log \left( 4 \left| B^{\pi^*}_{s,a} \right| / \delta \right)}{N}} \left( \sqrt{Var_{s,a}(\widehat{V}^{\pi^*}_h)} + \sqrt{Var_{s,a} \left( \widetilde{V}^{\pi^*}_{h,u} - \widehat{V}^{\pi^*}_h \right)} \right)$$

$$+ \frac{2 \log \left(4 \left|B^{\pi^*}_{s,a}\right| /\delta\right) (H-h)}{3N} + \left\|\widehat{V}^{\pi^*}_h - \widetilde{V}^{\pi^*}_{h,u}\right\|_\infty$$

$$\overset{(d)}{\leq} \sqrt{\frac{2 \log \left(4 \left|B^{\pi^*}_{s,a}\right| /\delta\right)}{N}} \sqrt{Var_{s,a}(\widehat{V}^{\pi^*}_h)} + \frac{2 \log \left(4 \left|B^{\pi^*}_{s,a}\right| /\delta\right)(H-h)}{3N}$$

$$+ \left\|\widehat{V}^{\pi^*}_h - \widetilde{V}^{\pi^*}_{h,u}\right\|_\infty \left(1 + \sqrt{\frac{2 \log \left(4 \left|B^{\pi}_{s,a}\right| /\delta\right)}{N}}\right)$$

$$\overset{(e)}{\leq} \sqrt{\frac{2 \log \left(4 \left|B^{\pi^*}_{s,a}\right| /\delta\right)}{N}} \sqrt{Var_{s,a}(\widehat{V}^{\pi^*}_h)} + \frac{2 \log \left(4 \left|B^{\pi^*}_{s,a}\right| /\delta\right)(H-h)}{3N}$$

$$+ \left\|u^{\pi^*} - u\right\|_\infty (H-h) \left(1 + \sqrt{\frac{2 \log \left(4 \left|B^{\pi^*}_{s,a}\right| /\delta\right)}{N}}\right).$$

(a) is due to triangle inequality, (b) is from Lemma 20, (c) is from Lemma 5, (d) is due to the fact that $\sqrt{Var(V)} \leq V$ and (e) is from Lemma 19. As this equality holds for all $u \in B^{\pi^*}_{s,a}$, we can take minimum in the RHS, which proves the first claim. The second claims can proved in the same manner. $\qquad\square$

**Lemma 22.** *For any given $\epsilon$ and all $(s,a) \in \mathcal{K}$, with probability larger than $1 - 2H\delta$,*

$$\left|\left(P(s,a) - \widehat{P}(s,a)\right)\widehat{V}^{\pi^*}_h\right| \leq \sqrt{\frac{2H \log(32KH^3/\delta\epsilon)}{N}} \sqrt{Var_{s,a}(\widehat{V}^{\pi^*}_h)} + \frac{2H^2 \log(32KH^3/\delta\epsilon)}{3N}$$

$$+ \left(\sqrt{\frac{2H \log(32KH^3/\delta\epsilon)}{N}} + 1\right) \frac{\epsilon}{4H},$$

$$\left|\left(P(s,a) - \widehat{P}(s,a)\right)\widehat{V}^{*}_h\right| \leq \sqrt{\frac{2H \log(32KH^3/\delta\epsilon)}{N}} \sqrt{Var_{s,a}(\widehat{V}^{*}_h)} + \frac{2H^2 \log(32KH^3/\delta\epsilon)}{3N}$$

$$+ \left(\sqrt{\frac{2H \log(32KH^3/\delta\epsilon)}{N}} + 1\right) \frac{\epsilon}{4H}.$$

*For simplicity, we set $c = c(\delta, H, \epsilon) = 2\log(64KH^4/\delta\epsilon)$.*

*Proof.* We set $B^{\pi^*}_{s,a}$ to be the evenly spaced elements in the interval $U^{\pi^*}_{s,a} \cap [-H,H]^H$ and $\left|B^{\pi^*}_{s,a}\right| = \left(\frac{4H^3}{\epsilon}\right)^H$. Then for any $u' \in U^{\pi^*}_{s,a} \cap [-H,H]^H$, we have $\min_{u \in B^{\pi^*}_{s,a}} \|u' - u\|_\infty \leq \epsilon/4H^2$. Note that $u^{\pi^*} \in U^*_{s,a} \cap [-H,H]^H$. Combining this with Lemma 21 implies the result. Similarly we can prove it for $B^*_{s,a}$. $\qquad\square$

**Lemma 23.** *For any given $\epsilon$ and all $(s,a) \in \mathcal{K}$, with probability larger than $1 - 2H\delta$,*

$$\left|\left(P(s,a) - \widehat{P}(s,a)\right)\widehat{V}^{\pi^*}_h\right| \leq \sqrt{\frac{2 \min\{K, |\mathcal{S}|\} \log(32KH^3/\delta\epsilon)}{N}} \sqrt{Var_{s,a}(\widehat{V}^{\pi^*}_h)}$$

$$+ \frac{2 \min\{K, |\mathcal{S}|\} H \log(32KH^3/\delta\epsilon)}{3N} + \left(\sqrt{\frac{2 \min\{K, |\mathcal{S}|\} \log(32KH^3/\delta\epsilon)}{N}} + 1\right) \frac{\epsilon}{4H},$$

$$\left|\left(P(s,a) - \widehat{P}(s,a)\right)\widehat{V}^{*}_h\right| \leq \sqrt{\frac{2 \min\{K, |\mathcal{S}|\} \log(32KH^3/\delta\epsilon)}{N}} \sqrt{Var_{s,a}(\widehat{V}^{*}_h)}$$

$$+ \frac{2 \min\{K, |\mathcal{S}|\} H \log(32KH^3/\delta\epsilon)}{3N} + \left(\sqrt{\frac{2 \min\{K, |\mathcal{S}|\} \log(32KH^3/\delta\epsilon)}{N}} + 1\right) \frac{\epsilon}{4H}.$$

*Proof.* Lemma 22 is proved by constructing a $\epsilon/4H$-net on $\widehat{V}_h^{\pi^*}$ via auxiliary MDP. Note that

$$\widehat{V}_h^* = \max_a \widehat{Q}_h^* = \max_a (r + \Phi \widehat{P}_{\mathcal{K}} \widehat{V}_{h+1}^*),$$

which means $\widehat{V}_h^*$ lies in a $K$-dimensional manifold in $[0, H]^{\mathcal{S}}$. We can make an $\epsilon/4H$-net on this manifold with $O(\frac{4H^3}{\epsilon})^{\min\{K, |\mathcal{S}|\}}$ points. With similar analysis as Lemma 22, we can prove this claim. $\qquad\square$

*Proof of Theorem 3.* From Lemma 15, with probability larger than $1 - \delta$, we have

$$\left\| Q_0^* - Q_0^{\widehat{\pi}} \right\|_\infty \le \left\| Q_0^* - \widehat{Q}_0^{\pi^*} \right\|_\infty + \left\| \widehat{Q}_0^{\widehat{\pi}} - Q_0^{\widehat{\pi}} \right\|_\infty + \epsilon_{\mathrm{PS}}$$

$$\le \frac{1}{1-\alpha} \left[ 2\left( \sqrt{\frac{cH^3 \min\{H, K, |\mathcal{S}|\}}{N}} + \frac{cH^2 \min\{H, K, |\mathcal{S}|\}}{3N} \right. \right.$$

$$\left. \left. + \left( \sqrt{\frac{c}{N}} + 1 \right) \frac{\epsilon}{4} \right) + \left( \sqrt{\frac{c}{N}} + 1 \right) \epsilon_{\mathrm{PS}} H \right] + \epsilon_{\mathrm{PS}},$$

where $\alpha = \alpha(\delta, H, \epsilon, N) = \sqrt{\frac{c(\delta, H, \epsilon)H^2}{N}}$ and the second inequality can be derived as in DMDP from Lemma 22 and Lemma 23. For $N \ge \frac{C \log(CKH\delta^{-1}\epsilon^{-1})H^3 \min\{H,K,|\mathcal{S}|\}}{\epsilon^2}$ with proper constant $C$, we have $\frac{1}{1-\alpha}(\sqrt{\frac{c}{N}} + 1) \le 2$, thus we have

$$\left\| V_0^* - V_0^{\widehat{\pi}} \right\|_\infty \le \epsilon + 3\epsilon_{\mathrm{PS}} H.$$

$\qquad\square$

## C.3 Sample Complexity for 2-TBSG

The value concentration becomes a little more tricky in 2-TBSG. The proof is similar to the case for DMDP, which only differs in the attendance of counter policy. The counter policy in 2-TBSG can be seen as the optimal policy in a DMDP induced by the policy of the opponent.

**Definition 4.** (Auxiliary Model) For a estimated transition model $\widehat{\mathcal{M}} = (\mathcal{S}_1, \mathcal{S}_2, \mathcal{A}, \widehat{P} = \Phi \widehat{P}_{\mathcal{K}}, r, \gamma)$ and a given anchor state pair $(s, a)$, the auxiliary transition model is $\widetilde{\mathcal{M}}_{s,a,u} = (\mathcal{S}_1, \mathcal{S}_2, \mathcal{A}, \widetilde{P} = \Phi \widetilde{P}_K, r + u\Phi^{s,a}, \gamma)$, where

$$\widetilde{P}_{\mathcal{K}}(s', a') = \begin{cases} \widehat{P}(s', a') & \text{if } (s', a') \ne (s, a) \\ P(s, a) & \text{otherwise,} \end{cases}$$

$\Phi^{s,a}$ is the column of $\Phi$ that corresponds to anchor state $(s, a)$ and $u$ is a variable that will be determined latter.

**Notations** For simplicity, we ignore $(s, a)$ in functions of auxiliary transition model $\mathcal{M}_{s,a,u}$. $c(\pi_1), \widehat{c}(\pi_1), \widetilde{c}_u(\pi_1)$ are the counter policies for $\pi_1$ in $\mathcal{M}, \widehat{\mathcal{M}}, \widetilde{\mathcal{M}}_{s,a,u}$. When it is clear in the context, we use $c$ as the counter policy function for $\pi_2$ as well. $\pi^* = (\pi_1^*, \pi_2^*), \widehat{\pi}^* = (\widehat{\pi}_1^*, \widehat{\pi}_2^*), \widetilde{\pi}^* = (\widetilde{\pi}_1^*, \widetilde{\pi}_2^*)$ are the equilibrium policies in $\mathcal{M}, \widehat{\mathcal{M}}, \widetilde{\mathcal{M}}_{s,a,u}$. We use $\widetilde{V}_u^\pi, \widetilde{Q}_u^\pi$ to denote value function and Q-function and $\widetilde{\pi}_u$ to be the optimal policy in $\mathcal{M}_{s,a,u}$.

**Definition 5.** (Feasible Set for $u$) For the auxiliary transition model $\widetilde{M}$, $U_{s,a}^\pi$ is defined as the set of $u$ so that $\widetilde{V}_u^\pi \in [0, 1/(1-\gamma)]^{\mathcal{S}}$ and $U_{s,a}^*$ is defined as the set of $u$ so that $\widetilde{V}_u^* \in [0, 1/(1-\gamma)]^{\mathcal{S}}$.

**Lemma 24.**

$$V^{c(\pi_2), \pi_2} \ge V^{\pi_1^*, \pi_2^*}, \quad V^{\pi_1, c(\pi_1)} \le V^{\pi_1^*, \pi_2^*}.$$

**Lemma 25.**
$$Q^{\pi_1, c(\pi_1)}(s, a) \geq Q^{\pi_1, c(\pi_1)}(s, c(\pi_1)(s)), \ \forall s \in \mathcal{S}_2$$
$$Q^{\pi_1^*, \pi_2^*}(s, a) \begin{cases} \geq Q^{\pi_1^*, \pi_2^*}(s, \pi_2^*(s)) & \forall s \in \mathcal{S}_2 \\ \leq Q^{\pi_1^*, \pi_2^*}(s, \pi_1^*(s)) & \forall s \in \mathcal{S}_1. \end{cases}$$

*These two equalities are also the sufficient condition for counter policy and equilibrium policy.*

*Proof.* The proof of Lemma 24 and 25 can be found in [2]. □

**Lemma 26.** *Let $\widehat{\pi} = (\widehat{\pi}_1, \widehat{\pi}_2)$ be a $\epsilon_{\mathrm{PS}}$-optimal policy in $\widehat{M}$.*

$$- \left( \left\| Q^{\widehat{c}(\pi_2^*), \pi_2^*} - \widehat{Q}^{\widehat{c}(\pi_2^*), \pi_2^*} \right\|_\infty + \left\| Q^{\widehat{\pi}_1, \widehat{\pi}_2} - \widehat{Q}^{\widehat{\pi}_1, \widehat{\pi}_2} \right\|_\infty + \epsilon_{\mathrm{PS}} \right) \mathbf{1} \leq Q^* - Q^{\widehat{\pi}_1, \widehat{\pi}_2}$$
$$\leq \left( \left\| Q^{\pi_1^*, \widehat{c}(\pi_1^*)} - \widehat{Q}^{\pi_1^*, \widehat{c}(\pi_1^*)} \right\|_\infty + \left\| Q^{\widehat{\pi}_1, \widehat{\pi}_2} - \widehat{Q}^{\widehat{\pi}_1, \widehat{\pi}_2} \right\|_\infty + \epsilon_{\mathrm{PS}} \right) \mathbf{1}.$$

*Proof.* We prove the second inequality and the first one can be proved by symmetry.

$$Q^* - Q^{\widehat{\pi}_1, \widehat{\pi}_2}$$
$$= Q^* - Q^{\pi_1^*, \widehat{c}(\pi_1^*)} + Q^{\pi_1^*, \widehat{c}(\pi_1^*)} - \widehat{Q}^{\pi_1^*, \widehat{c}(\pi_1^*)} + \widehat{Q}^{\pi_1^*, \widehat{c}(\pi_1^*)} - \widehat{Q}^* + \widehat{Q}^* - \widehat{Q}^{\widehat{\pi}} + Q^{\widehat{\pi}} - \widehat{Q}^{\widehat{\pi}}$$
$$\leq Q^{\pi_1^*, \widehat{c}(\pi_1^*)} - \widehat{Q}^{\pi_1^*, \widehat{c}(\pi_1^*)} + \widehat{Q}^* - \widehat{Q}^{\widehat{\pi}} + Q^{\widehat{\pi}} - \widehat{Q}^{\widehat{\pi}}$$
$$\leq \left( \left\| Q^{\pi_1^*, \widehat{c}(\pi_1^*)} - \widehat{Q}^{\pi_1^*, \widehat{c}(\pi_1^*)} \right\|_\infty + \left\| Q^{\widehat{\pi}_1, \widehat{\pi}_2} - \widehat{Q}^{\widehat{\pi}_1, \widehat{\pi}_2} \right\|_\infty + \epsilon_{\mathrm{PS}} \right) \mathbf{1}$$
□

**Lemma 27.** *Let $u^{\pi_1} = \gamma \left( \widehat{P}(s, a) - P(s, a) \right) \widehat{V}^{\pi_1, \widehat{c}(\pi_1)}$, $u^{\pi_2} = \gamma \left( \widehat{P}(s, a) - P(s, a) \right) \widehat{V}^{\widehat{c}(\pi_2), \pi_2}$, $u^* = \gamma \left( \widehat{P}(s, a) - P(s, a) \right) \widehat{V}^{\widehat{\pi}_1^*, \widehat{\pi}_2^*}$, we have*

$$\widehat{Q}^{\pi_1, \widehat{c}(\pi_1)} = \widetilde{Q}_{u^{\pi_1}}^{\pi_1, \widehat{c}(\pi_1)} = \widetilde{Q}_{u^{\pi_1}}^{\pi_1, \widetilde{c}(\pi_1)}, \widehat{Q}^{\widehat{c}(\pi_2), \pi_2} = \widetilde{Q}_{u^{\pi_2}}^{\widehat{c}(\pi_2), \pi_2} = \widetilde{Q}_{u^{\pi_2}}^{\widetilde{c}(\pi_2), \pi_2}, \widehat{Q}^* = \widetilde{Q}_{u^*}^{\widehat{\pi}_1^*, \widehat{\pi}_2^*} = \widetilde{Q}_{u^*}^*.$$

*Proof.* The proof of the first equality in two arguments are identical to Lemma 6. Combining with Lemma 25, we have the second equality. □

**Lemma 28.**
$$\left\| \widetilde{Q}_{u_1}^{\pi_1, \widetilde{c}_{u_1}(\pi_1)} - \widetilde{Q}_{u_2}^{\pi_1, \widetilde{c}_{u_2}(\pi_1)} \right\| \leq |u_1 - u_2| \frac{1}{1 - \gamma},$$
$$\left\| \widetilde{Q}_{u_1}^* - \widetilde{Q}_{u_2}^* \right\| \leq |u_1 - u_2| \frac{1}{1 - \gamma}$$

*Proof.* The proof is identical to Lemma 7. □

**Lemma 29.** *With probability larger than $1 - \delta$, and $\widehat{\pi}$ is a $\epsilon_{\mathrm{PS}}$-optimal policy in $\widehat{M}$.*

$$\left\| Q^{\pi_1^*, \widehat{c}(\pi_1^*)} - \widehat{Q}^{\pi_1^*, \widehat{c}(\pi_1^*)} \right\|_\infty \leq \frac{\gamma}{1 - \alpha} \left( \sqrt{\frac{c}{N(1 - \gamma)^3}} + \frac{c}{(1 - \gamma)^2 3N} + \left( \sqrt{\frac{c}{N}} + 1 \right) \frac{\epsilon}{4} \right),$$

$$\left\| Q^{\widehat{\pi}} - \widehat{Q}^{\widehat{\pi}} \right\|_\infty \leq \frac{\gamma}{1 - \alpha} \left( \sqrt{\frac{c}{N(1 - \gamma)^3}} + \frac{c}{(1 - \gamma)^2 3N} + \left( \sqrt{\frac{c}{N}} + 1 \right) \left( \frac{\epsilon}{4} + \frac{\epsilon_{\mathrm{PS}}}{1 - \gamma} \right) \right).$$

*where $c$ and $\alpha$ is defined as in Lemma 10 and Lemma 12.*

*Proof.* The proof is identical to Lemma 12 as we have Lemma 27 and Lemma 28 in 2-TBSG, which is the counterpart of Lemma 6 and Lemma 7. □

*Proof of Theorem 4.* From Lemma 26, with probability larger than $1 - \delta$, we have

$$
\begin{aligned}
&Q^* - Q^{\widehat{\pi}_1, \widehat{\pi}_2} \\
&\leq \left( \left\| Q^{\pi_1^*, \widehat{c}(\pi_1^*)} - \widehat{Q}^{\pi_1^*, \widehat{c}(\pi_1^*)} \right\|_\infty + \left\| \widehat{Q}^{\widehat{\pi}} - Q^{\widehat{\pi}} \right\|_\infty + \epsilon_{\text{PS}} \right) \mathbf{1} \\
&\leq \left( \frac{\gamma}{1-\alpha} \left[ 2 \left( \sqrt{\frac{c}{N(1-\gamma)^3}} + \frac{c}{(1-\gamma)^2 3N} + \left( \sqrt{\frac{c}{N}} + 1 \right) \frac{\epsilon}{4} \right) + \left( \sqrt{\frac{c}{N}} + 1 \right) \frac{\epsilon_{\text{PS}}}{1-\gamma} \right] \right. \\
&\quad \left. + \epsilon_{\text{PS}} \right) \mathbf{1}
\end{aligned}
$$

For $N \geq \frac{C \log(CK(1-\gamma)^{-1} \delta^{-1} \epsilon^{-1})}{(1-\gamma)^3 \epsilon^2}$ with proper constant $C$, we have $\frac{\gamma}{1-\alpha}(\sqrt{\frac{c}{N}} + 1) \leq 2$, thus

$$
V^* - V^{\widehat{\pi}_1, \widehat{\pi}_2} \leq (\epsilon + \frac{3\epsilon_{\text{PS}}}{1-\gamma}) \mathbf{1}.
$$

By symmetry, we have

$$
V^* - V^{\widehat{\pi}_1, \widehat{\pi}_2} \geq -(\epsilon + \frac{3\epsilon_{\text{PS}}}{1-\gamma}) \mathbf{1}.
$$

Thus we have

$$
\left\| V^* - V^{\widehat{\pi}_1, \widehat{\pi}_2} \right\|_\infty \leq \epsilon + \frac{3\epsilon_{\text{PS}}}{1-\gamma},
$$

which completes the proof.

$\square$

# D  Sample Complexity in General Linear Case

In the general linear case, the empirical MDP can be a pseudo MDP and hence we cannot get the optimal policy in the empirical MDP. Even so, the value iteration solver can still be applied to the pseudo MDP and we prove that it is a sample efficient algorithm. The proof relies on an observation that a discounted MDP can be approximated by a discounted finite horizon MDP with horizon $H = O(1/(1-\gamma) \log \epsilon^{-1})$. We use $\widehat{V}_h^*$ to represent the result of operating value iteration for $H - h$ steps in the empirical model and $V_h^*$ to be the result in true model. The initial value is $\widehat{V}_H^* = V_H^* = 0$. We use a similar analysis in FHMDP, but the pseudo MDP leads to some defect of previous proof and we revised the bound.

**Lemma 30.** *The error of performing value iteration for $H$ steps can be bounded:*

$$
\left\| \widehat{Q}_0^* - Q_0^* \right\|_\infty \leq \sum_{h=0}^{H-1} \gamma^{h+1} L \left\| \left( \widehat{P}_\mathcal{K} - P_\mathcal{K} \right) \widehat{V}_{h+1}^* \right\|_\infty.
$$

*Proof.* The proof is from iteratively using the following inequality.

$$
\begin{aligned}
\left\| \widehat{Q}_h^* - Q_h^* \right\|_\infty &= \left\| \left( r + \gamma \widehat{P} \widehat{V}_{h+1}^* \right) - \left( r + \gamma P V_{h+1}^* \right) \right\|_\infty \\
&\leq \left\| \gamma P \left( \widehat{V}_{h+1}^* - V_{h+1}^* \right) \right\|_\infty + \left\| \gamma \left( \widehat{P} - P \right) \widehat{V}_{h+1}^* \right\|_\infty \\
&\leq \gamma \left\| \widehat{Q}_{h+1}^* - Q_{h+1}^* \right\|_\infty + \gamma \left\| \left( \widehat{P} - P \right) \widehat{V}_{h+1}^* \right\|_\infty \\
&= \gamma \left\| \widehat{Q}_{h+1}^* - Q_{h+1}^* \right\|_\infty + \gamma \left\| \Phi \left( \widehat{P}_\mathcal{K} - P_\mathcal{K} \right) \widehat{V}_{h+1}^* \right\|_\infty \\
&\leq \gamma \left\| \widehat{Q}_{h+1}^* - Q_{h+1}^* \right\|_\infty + \gamma L \left\| \left( \widehat{P}_\mathcal{K} - P_\mathcal{K} \right) \widehat{V}_{h+1}^* \right\|_\infty.
\end{aligned}
$$

$\square$

The auxiliary MDP technique can be used in the same way to analyze $(\widehat{P} - P)\widehat{V}^*_{h+1}$ as in Lemma 22. Here we give two lemmas, which are counterparts of Lemma 19 and Lemma 20 for FHMDP. Note that the total variance technique is not applicable in pseudo MDP as Lemma 3 no longer holds. Therefore, we use Hoeffding's inequality to analyze the concentration.

**Lemma 31.** *For all $u, u' \in \mathbb{R}^H$ and policy $\pi$,*

$$\left\| \widetilde{Q}^*_{h,u} - \widetilde{Q}^*_{h,u'} \right\|_\infty \leq (H-h)L^{H-h}\|u - u'\|_\infty.$$

*Proof.* Similar to the proof of Lemma 19, we have

$$
\begin{aligned}
\left\| \widetilde{Q}^\pi_{h,u} - \widetilde{Q}^\pi_{h,u'} \right\|_\infty &= \left\| \left( r + u_h \Phi^{s,a} + \widetilde{P}V^\pi_{h,u} \right) - \left( r + u'_h \Phi^{s,a} + \widetilde{P}V^\pi_{h,u'} \right) \right\|_\infty \\
&\leq \|(u_h - u'_h)\Phi^{s,a}\|_\infty + \left\| \widetilde{P}^\pi \left( \widetilde{Q}^\pi_{h+1,u} - \widetilde{Q}^\pi_{h+1,u'} \right) \right\|_\infty \\
&= \|(u_h - u'_h)\Phi^{s,a}\|_\infty + \left\| \Phi \widetilde{P}^\pi_{\mathcal{K}} \left( \widetilde{Q}^\pi_{h+1,u} - \widetilde{Q}^\pi_{h+1,u'} \right) \right\|_\infty \\
&\leq |u_h - u'_h| + L \left\| \widetilde{Q}^\pi_{h+1,u} - \widetilde{Q}^\pi_{h+1,u'} \right\|_\infty \\
&\leq \sum_{i=h}^{H-1} L^{i-h} |u_i - u'_i| \\
&\leq (H-h)L^{H-h}\|u - u'\|_\infty.
\end{aligned}
$$

$\square$

**Lemma 32.** *For a given finite set $B^*_{s,a} \subset U^*_{s,a} \cap [-H, H]^H$ and $\delta \geq 0$, with probability greater than $1 - \delta$, it holds for all $u \in B^*_{s,a}$ that*

$$\left| \left( P(s,a) - \widehat{P}(s,a) \right) \widetilde{V}^*_{h,u} \right| \leq \sqrt{\frac{H^2 \log\left(2\left|B^*_{s,a}\right|/\delta\right)}{2N}}.$$

*Proof.* This is the direct application of Hoeffding's inequality as $\widetilde{V}^\pi_{h,u}$ and $\widetilde{V}^*_{h,u}$ is independent of $\widehat{P}(s,a)$. $\square$

**Lemma 33.** *For a given finite set $B^{\pi^*}_{s,a} \subset U^{\pi^*}_{s,a} \cap [-H, H]^H$ and $B^*_{s,a} \subset U^*_{s,a} \cap [-H, H]^H$ and $\delta \geq 0$, with probability greater than $1 - H\delta$, it holds for all $u \in B^*_{s,a}, h \in [H]$ that*

$$\left| \left( P(s,a) - \widehat{P}(s,a) \right) \widehat{V}^*_h \right| \leq \sqrt{\frac{H^2 \log\left(2\left|B^*_{s,a}\right|/\delta\right)}{2N}} + \min_{u \in B^*_{s,a}} \|u^* - u\|_\infty (H-h)L^{H-h}.$$

*Proof.* Combining Lemma 31 and Lemma 32, we have

$$
\begin{aligned}
\left| \left( P(s,a) - \widehat{P}(s,a) \right) \widehat{V}^*_h \right| &= \left| \left( P(s,a) - \widehat{P}(s,a) \right) \left( \widetilde{V}^*_{h,u} + \widehat{V}^*_h - \widetilde{V}^*_{h,u} \right) \right| \\
&\leq \left| \left( P(s,a) - \widehat{P}(s,a) \right) \widetilde{V}^*_{h,u} \right| + \left\| \widehat{V}^*_h - \widetilde{V}^*_{h,u} \right\|_\infty \\
&\leq \sqrt{\frac{H^2 \log\left(2\left|B^*_{s,a}\right|/\delta\right)}{2N}} + \|u^* - u\|_\infty (H-h)L^{H-h}.
\end{aligned}
$$

As this equality holds for all $u \in B^{\pi^*}_{s,a}$, we can take minimum in the RHS, which proves the first claim. $\square$

**Lemma 34.** *For any given $\epsilon$ and all $(s, a) \in \mathcal{K}$, with probability larger than $1 - H\delta$,*

$$\left| \left( P(s,a) - \widehat{P}(s,a) \right) \widehat{V}_h^* \right| \leq \sqrt{\frac{H^4 \log(8H^3 L^2/\delta\epsilon)}{2N}} + \frac{\epsilon}{4HL}.$$

*Proof.* We set $B_{s,a}^{\pi^*}$ to be the evenly spaced elements in the interval $U_{s,a}^{\pi^*} \cap [-H, H]^H$ and $\left| B_{s,a}^{\pi^*} \right| = (\frac{4H^3 L^{2H}}{\epsilon})^H$. Then for any $u' \in U_{s,a}^{\pi^*} \cap [-H, H]^H$, we have $\min_{u \in B_{s,a}^{\pi^*}} \|u' - u\|_\infty \leq \epsilon/4H^2 L^{2H}$. Note that $u^{\pi^*} \in U_{s,a}^* \cap [-H, H]^H$. Combining this with Lemma 33 implies the result. $\quad\square$

*Proof of Theorem 5.* From Lemma 34, with probability larger than $1 - \delta$, we have

$$\begin{aligned}
\left\| \widehat{V}_0^* - V_0^* \right\|_\infty &\leq \left\| \widehat{Q}_0^* - Q_0^* \right\|_\infty \\
&\leq \sum_{h=0}^{H-1} \gamma^{h+1} L \left\| \left( \widehat{P}_\mathcal{K} - P_\mathcal{K} \right) \widehat{V}_{h+1}^* \right\|_\infty \\
&\leq \sum_{h=0}^{H-1} \gamma^{h+1} L \left( \sqrt{\frac{H^4 \log(8H^4 L^2/\delta\epsilon)}{2N}} + \frac{\epsilon}{4HL} \right) \\
&\leq \sqrt{\frac{H^6 L^2 \log(8H^3 L^2/\delta\epsilon)}{2N}} + \frac{\epsilon}{4}
\end{aligned}$$

Choosing $N \geq CH^6 L^2 \log(CKHL/\delta\epsilon)(\epsilon(1 - \gamma))^{-2}$, we have $\left\| \widehat{V}_0^* - V_0^* \right\|_\infty \leq \epsilon(1 - \gamma)/2$. Now we replace $H$ with $O((1 - \gamma)^{-1} \log(1/\epsilon))$ and we have $\|V_0^* - V^*\|_\infty \leq \epsilon(1 - \gamma)/2$ by the convergence of value iteration. So we have $\left\| \widehat{V}_0^* - V^* \right\|_\infty \leq \epsilon(1 - \gamma)$, which implies the greedy policy with respect to $\widehat{V}_0^*$ is an $\epsilon$-greedy policy in the true MDP [5].

$\quad\square$