[Reviews · NeurIPS 2020]

Review 1

Summary and Contributions: The paper looks at the use of model-based methods for learning near-optimal policies in a linear (low-rank) MDP using a generative model. Specifically, under an anchor state assumption, the generative model is queried at the set of anchor states N times and the samples are used to build an empirical model. Using the empirical next state distribution over these anchor states, the model for all states in the MDP can be specified. As such, a planning algorithm is invoked to compute the optimal policy in the empirical MDP, hence, the title plug-in solver for feature based RL. A theoretical analysis of the sample complexity of learning an epsilon-optimal policy is given and shown to be minimax optimal. Overall, such results are presented for the discounted infinite horizon MDP, fixed horizon MDP and two-player turn-based stochastic game (TBSG). Update: Thanks for responding to the questions. The issue in the appendix was a typo and indeed easily fixable. Also, I now see that the despite the similarity of the algorithm and construction to previous work, there is a novel aspect to it. However, the nature still seems a bit incremental to me and there are similar questions about the assumption about anchor states and the requirement of that being a known set. As such, I would update my rating to a weak accept.

Strengths: The paper combines recent results in two separate threads of sample complexity in RL: model-based planning with generative model (AKY20) and learning in feature-based MDP under low-rank transitions (YW19). Overall, the paper is well written with required lemmata and propositions described appropriately in tandem with the main results. Specifically, the following results are shown: 1. Linear DMDP: Estimate p(.|s,a) on given anchor states. Use anchor state representation for building complete transition model. The algorithm is claimed to have optimal sample complexity where the rate was shown in YW19. 2. Finite horizon MDP: Similar result is shown with the same estimate for models. 3. TBSG: Same sample complexity result under a linear MDP assumption. Results follow in the same manner as DMDP. 4. Sample complexity for relaxed anchor state assumption: An updated sample complexity result is given based on literature on pseudo-MDPs. Dependence on 1/1-\gamma is different but no discussion is given on optimality. Thus, for linear MDPs with anchor state assumption, the paper presents many results on model-based planning with optimal sample complexity upto logarithmic factors. Further details below.

Weaknesses: Despite the near-optimal sample complexity bounds presented in the paper, the paper seems to fall short significantly on novelty and significance issue. Details below: Discussion on related work: The pitch of the paper is made in a way which suggests that there are no results on model-based RL when function approximation is used. This overplays the significance and novelty of the result. However, recently, there have been many papers which look at model-based algorithms: Wen et al 2019 (which is cited in the paper) is said to be a model-based method whereas it clearly studies model-based RL. If one looks at the corresponding LQR like problems, effectively all results are model-based. Pires and Szepesvari (COLT 2016) discuss policy error bounds in model based RL. As such, the related work discussion can be a bit misleading. Novelty: The paper looks at model-based planning and linear models. As such, all the required techniques and assumptions have already been looked at in both contexts in previous papers. For instance, YW19 discuss the anchor state assumption in detail and propose optimal-bounds for a value-iteration algorithm. On the other hand, the key ideas for model-based planning have completely borrowed from AKY20 with the all lemmata exactly the same as in the tabular case. Further, the discussion about the Q-function lying on a one-dimensional manifold and the epsilon-net should be discussed in reference to AKY20 as these are not new results. For extending results to linear MDPs, the anchor state assumptions are critical and how they manifest in typical dynamic programming methods has been studied in YW19. Can the authors comment on the specific challenge in the analysis and the novelty component in the paper? Also, in the appendix, intermediate lemma should be clearly distinguished as a known/adapted result to a new result. As such, there is no significant novelty in the technical component. Model-based planning: For a feature-based model, using function approximation is quite critical as the paper discusses. However, for the plug-in solver or the overall approach in general, nothing is said about the computational aspect of planning. For instance, the model representation itself is now count-based which seems unappealing for continuous state spaces. If further details are added on how these models can be used and how these differ significantly from empirical QVI approaches would improve the paper. Anchor states and pseudo-MDP: It is known that the number of anchor states can be exponential in dimension of features. This would further worsen the computational complexity aspect. For the general linear transition model case, it is claimed that the value iteration methods work. Can the authors cite the relevant results here? I'm under the impression that one would need the l-1 norm of \lambda vector to be <=1 for that to be true. Otherwise, the value iteration analogue would be the NeurIPS'19 paper by Zanette et al where the error propagation of LSVI/FQI algorithm is discussed. Further comparison and discussion about results from that paper would be helpful. Also, what is the optimal dependence on 1/(1-\gamma)? Potential incorrectness: Please see below for the question.

Correctness: To my best efforts, the analysis seems correct apart from one key step: In line 105 in the appendix (proof of lemma 6): Firstly, it would be better if the second inequality for \tilde(Q)^\pi derivation is expanded on with all missing steps. There is much addition subtraction going on here and needs clear writing. Mainly, in the next step, the first term clearly gives \hat{Q}^\pi. How is the next term in second equality 0? In effect, how is the third equality derived? This is a key step and if incorrect would invalidate the whole paper.

Clarity: Overall, the paper is well written apart from clear discussion of how this work differs from previous results and analysis approaches.

Relation to Prior Work: As described previously, attribution to previous work is a key issue with the current paper and needs improvement.

Reproducibility: Yes

Additional Feedback:


Review 2

Summary and Contributions: This paper studies the sample complexity problem in reinforcement learning. The authors show that a near-optimal solution to the MDP can be learned by a plug-in solver with O(K/((1-gamma)^3*eps^2)) samples, which can be done efficiently. The main result of the paper depends crucially on some assumptions, such as the linear model representation assumption and the anchor-state assumption.

Strengths: In many cases, when learning a solution to an MDP, the learner only has access to samples of how the states transition among one another. Thus the topic of sample complexity in the reinforcement learning domain is important and worth investigation. The paper is nicely written and easy to follow. The results look interesting and solid. The main result matches a lower bound in a previous lower bound, closing the gap in sample complexity in this setting.

Weaknesses: But of course, such results depends on some strong assumptions. They require that the MDP has a linear transition model that depends on features of states and actions. An other assumption is the anchor-state assumption. With such assumptions, the MDP problem becomes very well-structured, and thus the samples needed to learn a good solution can be greatly reduced. These assumptions would definitely make the results inapplicable to many problems. Another issue is that the authors assume that the learner has access to a generative model for state transition function. I am not sure if this is realistic.

Correctness: The claims and results look correct to me. But I did not check all the proof details.

Clarity: The paper is well-organized and nicely written.

Relation to Prior Work: The paper clearly discussed how it is related to and how it is different from previous works.

Reproducibility: Yes

Additional Feedback: To summarize, although the paper makes some strong assumptions, I think it is still a nice paper from a theoretic point of view. It is expected that the MDP is easier to solve when it has nice structures, and this paper can tell us how to make use of the linear structure of the MDP. Minor: 1. Line 45, page 2, "an near ..." -> "a near ..." 2. Line 169, page 5, "an probability ..." -> "a probability ..." After author response: The authors argue that the assumptions are not very strong because it is general. But being general does not make it weak. To achieve similar performances of non-linear models, we may need to add many more features, making the model complicated again. But I understand that analyzing a non-linear model could be much more challenging, and that may be why linear assumptions are commonly used. My score remains unchanged after reading other reviewers' comments and the author response.


Review 3

Summary and Contributions: This paper studies how to use plug-in solvers to find near-optimal policies for a linear MDP, which is an important model for studying the sample complexity of RL. This paper makes a significant contribution about how to construct an empirical model and apply plug-in solvers to find near-optimal policies. The result is surprising in the sense that although the complexity of the model is dS, model-based approaches learn a model using O~(d) samples and an plug-in solver can obtain a near-optimal policy based on this model. The paper also shows that this method is near-optimal under the anchor-state assumption.

Strengths: 1. The paper provides a nice result of model-based methods for linear MDPs. It shows that even for high complexity models, a low-complexity representation could produce a good policy. 2. For an interesting special case, the anchor-state assumption, the paper shows that the sample complexity of the plug-in solver approach is optimal. 3. The technique used in this paper is a non-trivial extension of the decoupling technique used in the tabular setting.

Weaknesses: The result requires access to a simulator. No experiments (ok for a thoery paper).

Correctness: The results seem correct to me.

Clarity: The paper is well written.

Relation to Prior Work: The difference between this work and previous contributions is clearly discussed.

Reproducibility: Yes

Additional Feedback: After reading the author's feedback and discussing with other reviewers, I would like to keep my original score unchanged.


Review 4

Summary and Contributions: ####### Post Rebuttal ####### After reading other reviews and the authors response, I decide to maintain my sore (7) ######################### The paper studies the sample complexity of finding $\epsilon$-near optimal policy in linear MDP given only access to a generative model. Under anchor states assumption, the authors show that planning in the empirical model via any plug-in solver achieves a minimax sample complexity O(K / (1-\gamma)^3 \epsilon^2) where K is the dimension of feature space. Moreover, the author show that value iteration in the empirical model could be as well efficient with more relaxed assumption but with worser dependency in the planning horizon 1/(1-\gamma)

Strengths: The paper proves minimax sample complexity of model-based method to solve Linear MDP in the generative model setting. The minimax sample complexity was already achieved by model free method in (Yang and Wang 2019) but with smaller range of epsilon (epsilon \in [0,1] instead of [0, 1/\sqrt{1-\gamma}] of this paper). In terms of proof techniques, the paper introduces a clever way to decouple the dependance between the empirical transition kernel and the value function of the empirical MDP. This was done using auxiliary MDPs. it seems to be inspired by absorbing MDPs technique used in tabular MDP recently (Agarwal et al 2019)

Weaknesses: I think the main limitation of the paper is the strong assumption. I am wondering if the authors could provide an example of MDP where the anchor states assumption holds even approximately.

Correctness: I didn't read the proofs in details but they seem correct to me.

Clarity: The paper is well written

Relation to Prior Work: The connection with related work is well discussed. I would encourage the author to summary in form of table (could be in appendix) all the sample complexity results with generative model of prior works and this works, highlighting the different assumptions used for each work.

Reproducibility: Yes

Additional Feedback: It is not clear in the main paper why we can assume that \phi(s, a) is a probability vector with loss of generality. it seems that linear MDP + anchor states implies that P is factorized by two probability matrices where the rows of first matrix are given by \lamda^{(s, a)}. I think this should be more reflected in the main paper. it would give also more intuition why the \hat{P} is an unbiased estimate of P. typos: l 32 nerual -> neural l 162 an direct -> a direct l 234 \bar{P}_K is not defined before


Review 5

Summary and Contributions: The paper consider model-based learning in linear MDPs with a generative model. Further it assumes that it has access to anchor states and all other features live in the convex hull of the anchor states's features. The algorithm uses generative model to get samples for each anchor state-action pairs, and then construct the entire model by using the known linear combination. The constructed transition is a valid transition matrix in this case. The theory shows that under this assumption, the algorithm achieves optimal sample complexity.

Strengths: The paper proposes a simple algorithm that achieves minimax sample complexity under the given assumption. This results extend previous generative model results from tabular setting to linear transition setting, and is the first such result in the literature as far as I can tell. Also, the proof of the results require an interesting and novel construction of a set auxiliary MDPs (one for each anchor state-action) which is critical for the follow up uniform convergence type analysis.

Weaknesses: The algorithm relies on the anchor state assumption which states that any state-action's feature can be represented by a linear combination of the features of anchor states. Further, it also assumes that the algorithm know the anchor states (the samples are collected at the anchor states). Such assumption seems quite strong to me and this is a much stronger assumption than linear MDP alone.

Correctness: I did not check the appendix carefully, but the claims and the algorithm look reasonable to me.

Clarity: I think the paper is well written.

Relation to Prior Work: It has thorough comparison to existing works.

Reproducibility: No

Additional Feedback:

[Author Response · NeurIPS 2020]

We thank all reviewers for their comments and insightful reviews. We will address the concerns as follows. Typos will
be corrected in the final version.
**Reviewer 1** Thank you for your comments. We hope following explanations can answer your questions.
**Related work** Model-based algorithms with function approximation have been indeed studied in prior works. What we
want to claim is that we are the first to study the *plug-in* solver approach with function approximation. We believe this
is a nontrivial contribution to the RL community.
**Novelty** Although we have applied the anchor-state model from [Yang and Wang] and our technique has certain
similarity to [Agarwal et al., 2019], we stress that neither of the two papers can be directly applied in our setting for a
plug-in approach of model-based RL with function approximation. In fact, our technique is highly non-trivial and
inspires several new understandings in terms of model-based RL with features. Moreover, our results can also be
applied to the setting without anchor-state by restricting the plug-in solver to value iteration. In addition, turn-based
stochastic game, a multi-agent extension of MDP, is analyzed by using the optimal response policy in the auxiliary
MDP to approximate empirical optimal response policy.
More specifically, the model-free algorithm in [Yang and Wang] relies on monotonicity and variance reduction, which
cannot be applied to model-based setting with a plug-in solver. The absorbing MDP in [Agarwal et al., 2019] cannot
be applied to linear MDP as it destroys the linear dependence, which inspired us to invent a *new auxiliary MDP*.
This auxiliary MDP fixes the transition of an anchor state-action and changes the entire transition kernel via linear
dependence (Definition 2). The auxiliary MDP technique we propose can also be applied to tabular MDP, which covers
absorbing MDP. Another critical disadvantage of the absorbing MDP in [Agarwal et al., 2019] is that it can only recover
state values, while our technique can recover the entire state-action values (Lemma 6).
**Model-based planning** First, we want to emphasize that arbitrary planning algorithm is suitable in our algorithm, so the
computational complexity of the planning algorithm is not our main focus. It is known that the exactly optimal policy
in MDP and TBSG can be obtained in strong polynomial time $\widetilde{O}(\text{poly}(|\mathcal{S}||\mathcal{A}|(1-\gamma)^{-1}))$ by policy iteration/strategy
iteration [Ye, 2011, Hansen et al., 2013]. Approximate dynamic programming methods like LSVI/FQI can utilize
the features to achieve $\widetilde{O}(\text{poly}(K(1-\gamma)^{-1}\epsilon^{-1}))$ computational complexity . In addition, one can use the learning
algorithm 'Optimal Phased Parametric Q-Learning' in [Yang and Wang] to do planning, which has computational
complexity of $\widetilde{O}(K(1-\gamma)^{-3}\epsilon^{-2})$ (i.e. same to the sample complexity result in our work and immediately achieves
minimax computational complexity).
**Anchor states and pseudo-MDP** The analysis in [Zanette et al., 2019] requires $\|\lambda\|_1 \leq 1 + \frac{1}{H}$ so that the
error will not amplify exponentially and fails when $\|\lambda\|_1$ is larger than this threshold. Our result shows that
empirical value iteration (EVI) is a sample efficient algorithm for bounded $\|\lambda\|_1$ with sample complexity
$\widetilde{O}(K \max_{s,a} \|\lambda(s,a)\|_1^2 \text{poly}((1-\gamma))\epsilon^{-2})$, which demonstrate that EVI is sample efficient for $\|\lambda\|_1 > 1 + \frac{1}{H}$. Note
that [Zanette et al., 2019] assumes linear representation of $Q^*$, which is different from our assumption. To our best
knowledge, the minimax sample complexity in linear MDP without anchor state assumption is still unknown.
**Correctness** We apologize for the typos in the appendix (proof of Lemma 6). The correct and more detailed version
is $\widetilde{Q}_{u^\pi}^\pi = (I - \gamma \widetilde{P}^\pi)^{-1}(r + \Phi^{s,a}u^\pi) = (I - \gamma \widetilde{P}^\pi)^{-1}(r + \Phi^{s,a}\gamma(\widehat{P}(s,a) - P(s,a))\widehat{V}^\pi) = (I - \gamma \widetilde{P}^\pi)^{-1}((I -$
$\gamma \widehat{P}^\pi)\widehat{Q}^\pi + \gamma \Phi(\widehat{P}_\mathcal{K} - \widetilde{P}_\mathcal{K})\widehat{V}^\pi) = (I - \gamma \widetilde{P}^\pi)^{-1}((I - \gamma \widehat{P}^\pi)\widehat{Q}^\pi + \gamma(\widehat{P} - \widetilde{P})\widehat{V}^\pi) = (I - \gamma \widetilde{P}^\pi)^{-1}(I - \gamma \widetilde{P}^\pi)\widehat{Q}^\pi = \widehat{Q}^\pi$.
Note that $\widehat{P}_\mathcal{K} - \widetilde{P}_\mathcal{K}$ has all zero rows except row $(s,a)$ by the definition of auxiliary MDP.
**Reviewer 2 & Reviewer 3 & Reviewer 4 & Reviewer 5** Thank you for your appreciation. We will fix typos and
clarify some of the confusions in the next version. Below, we address the common concern about the assumption in this
paper.
**Strong Assumptions** Our assumptions on linear MDP are widely used in literature as discussed in Section 4. Anchor
state assumption indeed appears a strong assumption, however this is rather general: this assumption essentially
assumes all the features vectors lie in a convex hull, which is without loss of generality. The number of vertices of the
convex hull is the number of anchor-state-action pairs. The number of vertices can be small in many cases (see e.g.,
[Blum et al., 2019] "Sparse approximation via generating point sets" and reference therein). Moreover, our results also
apply to the approximate model setting (Theorem 2).
Furthermore, we show that the anchor state assumption is essential to obtain an eligible empirical model by showing an
hard instance (Proposition 3). Our work also gives a minimax sample complexity algorithm for $\|\lambda\|_1 = 1$ (anchor state
assumption) and efficient algorithm for $\|\lambda\|_1 > 1$ but bounded.
Generative model is a meaningful oracle which receives much attention (see Section 2 for a detailed review) as it
separates the subtle exploration questions from learning. In many realistic settings we also have a simulator to generate
samples from arbitrary state-action pair. For instance, learning in physical simulators allows this kind of sampling.
Moreover, games like Go and chess are turn-based stochastic game that can be viewed as generative model.
**Table of previous results (R4)** Due to limited space, we cannot put the table in this rebuttal. A table of previous results
will be added in the final submission. We will also move some discussion of estimating the transition kernel $P$ in the
appendix to the main paper.
58

[Meta-Review · NeurIPS 2020]

The paper provides nice near-optimal sample complexity results for a setting of feature-based MBRL. The results are nontrivial extensions of previous tabular results. On the other hand, it requires a pretty strong anchor-state assumption, which to some extent limits the significance of the results.